# Iron-sulfur clusters in SARS-CoV-2 exoribonuclease and methyltransferase complexes: relevance for viral genome proofreading and capping

Nunziata Maio [1] ✉, Umberto Terranova[2], Yan Li [3], J. Martin Bollinger Jr. [4,5], Carsten Krebs[4,5] & Tracey A. Rouault [1] ✉

Coronaviruses rely on a multifunctional replication-transcription complex to ensure genome fidelity and support viral propagation. Within this complex, the nsp14-nsp10 heterodimer possesses 3′−5′ exoribonuclease (ExoN) activity, while nsp14 alone functions as an N7-methyltransferase and the nsp16/nsp10 complex completes viral RNA capping via its 2′-O-methyltransferase. Here, we report that nsp14 and nsp10 ligate [$Fe_4S_4$] clusters when purified anoxically, in sites previously modeled as zinc centers. Quantum mechanics/molecular mechanics simulations revealed distinct reduction potentials for these iron-sulfur (Fe-S) clusters, and redox titrations demonstrated that changes in oxidation state modulate RNA binding by nsp14 and the nsp10/nsp16 complex. Functionally, Fe-S clusters enhance the methyltransferase activities of nsp14 and nsp10/nsp16, while leaving the ExoN activity unaffected. These findings uncover a redox-regulated role for Fe-S clusters in SARS-CoV-2 RNA processing and suggest that the viral core enzymatic functions may be modulated by the redox state of their Fe-S cofactors.

Coronaviruses, including SARS-CoV-2, the etiological agent of COVID-19, are a family of positive sense, single stranded RNA viruses with a genome of approximately 30 kb[1]. Encoded by the 5′ proximal two-thirds of the SARS-CoV-2 genome, two large polyproteins are translated by host ribosomes and subsequently cleaved by viral proteases to generate 16 nonstructural proteins[2,3] (nsps). These nsps primarily assemble into the replication and transcription complex (RTC) essential for the virus cycle. The RNA-dependent RNA polymerase (RdRp) is a core component of the RTC. The N-terminal domain of nsp12, the catalytic subunit of the RdRp, known as the nidovirus RdRp-associated nucleotidyltransferase (NiRAN) domain[4], has recently been shown to catalyze the formation of the core

cap structure GpppA-RNA[5] located at the 5′ end of the viral genome. Similar to eukaryotic mRNAs, the coronavirus genome contains a 5′ cap consisting of a methylated guanosine linked to the first nucleotide, typically adenosine (A), via a reverse 5′ to 5′ triphosphate bond[6,7]. Initially identified in vaccinia virus[8] and reovirus[9], the 5′ cap plays an essential role in RNA stability, initiation of translation, and protection from nucleases[10]. The canonical capping pathway, conserved across most viruses and eukaryotes[11,12], involves sequential enzymatic activities. In SARS-CoV-2, this pathway begins with the RNA triphosphatase activity of nsp13[13], which removes the γ-phosphate from nascent RNA (5′-pppRNA) to yield diphosphorylated RNA (5′-ppRNA). The NiRAN domain of nsp12 then

[1]Eunice Kennedy Shriver National Institute of Child Health and Human Development, National Institutes of Health, Bethesda, MD 20892, USA. [2]Faculty of Medicine and Health Sciences, The University of Buckingham, Buckingham MK18 1EG & Crewe Campus, Crewe Green Road, Crewe CW1 5DU, UK. [3]National Institute of Neurological Disorders and Stroke, National Institutes of Health, Proteomics Core Facility, Bethesda, MD 20892, USA. [4]Department of Chemistry, The Pennsylvania State University, University Park, PA 16802, USA. [5]Department of Biochemistry and Molecular Biology, The Pennsylvania State University, University Park, PA 16802, USA. ✉e-mail: nunziata.maio@nih.gov; rouault@mail.nih.gov

mediates the transfer of monophosphorylated RNA (5'-pRNA) onto nsp9, followed by its transfer to guanosine diphosphate (GDP) to form GpppA-RNA[5]. The final steps consist of the guanine-N7 methylation by nsp14[14] and 2'-O ribose methylation by the nsp16/nsp10 complex[15], yielding a mature 5' cap structure essential for the viral life cycle.

In addition to its N7-MTase activity, nsp14 has 3'−5' exoribonuclease (ExoN) activity associated with its N-terminal domain[16]. The proofreading activity of the ExoN domain allows nsp14 to remove misincorporated nucleotides, significantly enhancing the fidelity of viral RNA synthesis[17]. Notably, the ExoN activity of nsp14 is strongly stimulated by its interaction with nsp10, which increases its catalytic efficiency by up to 35-fold[18,19]. This proofreading function, unique among RNA viruses, enables coronaviruses to maintain their unusually large genome size while ensuring genomic stability[20].

Recently, the crystal structure of the SARS-CoV-2 nsp14 in the absence of nsp10 was determined at 2.5 Å resolution[21], complementing previous structures solved in complex with nsp10[22]. This comparison provides insights into the structural dynamics of nsp14, revealing how its N-terminal "lid" region undergoes significant conformational changes upon nsp10 binding. In the absence of nsp10, the N-terminal exoribonuclease domain of nsp14 exhibits structural heterogeneity, with large regions undefined by electron density[21] −suggesting that there is intrinsic flexibility. The N-terminal domain (residues 1-70) of nsp14 acts as a 'lid' that shields the nsp10 binding region and occludes the ExoN active site. Upon nsp10 binding, the lid undergoes substantial structural rearrangement that exposes the ExoN site, thereby explaining the allosteric mechanism by which nsp10 modulates nsp14 exonuclease activity.

We previously reported the presence of two [Fe$_4$S$_4$] clusters in the catalytic subunit, nsp12, of the RdRp[23] and of one [Fe$_4$S$_4$] cluster in the treble clef site within the zinc-binding domain (ZBD) of the helicase, nsp13[24]. The iron-sulfur (Fe-S) clusters in the RdRp and in nsp13 were bound in sites that had been modeled as zinc centers in available structures of the proteins purified under standard aerobic conditions[25–31], and were susceptible to oxidative degradation by a small molecule stable nitroxide, TEMPOL, both in vitro[23,24] and in vivo[32]. Our work has been building upon multiple prior reports demonstrating that zinc can replace native Fe-S cofactors during the aerobic purification of Fe-S proteins[33–37]. This substitution occurs due to cluster oxidation and degradation by oxygen and reactive oxygen species[38,39]. Notably, Fe-S clusters, essential inorganic cofactors commonly associated with biological redox reactions, have been identified in numerous DNA- and RNA- processing enzymes[35,40–47]. Within the SARS-CoV-2 RTC, the RdRp and helicase, both characterized as Fe-S proteins, function as RNA-processing enzymes. While the precise roles of Fe-S clusters in host and viral proteins have not yet been defined, these cofactors are generally critical for the proper function of the enzymes in which they are found.

These unexpected discoveries of Fe-S clusters in RNA-processing enzymes within the SARS-CoV-2 RTC led us to investigate two additional components, nsp14 and nsp10. Both proteins contain zinc-binding motifs typically associated with metal coordination, raising the possibility that they may also ligate Fe-S clusters.

## Results

### SARS-CoV-2 nsp14 and nsp10 recruit the host Fe-S biogenesis components for cluster acquisition

Nsp14 is a bifunctional protein with three metal-binding sites present in all available structures[21,22]. Two sites have Cys$_3$His and one has Cys$_2$His$_2$ ligand sets. These ligands are all completely conserved among the seven human pathogenic coronaviruses (Supplementary Fig. 1a). The protein also features a LYR (leucine-tyrosine-arginine)-like motif (Supplementary Fig. 1a), previously characterized as a potential binding site for the cochaperone HSC20 (also known as HSCB) of the Fe-S cluster biogenesis machinery[23,24,48–51]. HSC20 enhances the ATPase activity of its cognate chaperone, HSPA9 in mammalian cells, facilitating the transfer of

Fe-S clusters from the primary scaffold ISCU (iron-sulfur cluster assembly scaffold) to recipient proteins, while also ensuring selective incorporation of Fe-S clusters into appropriate clients via direct binding to specific amino acid sequences, such as LYR-like motifs, in recipient Fe-S apo-proteins[48,52]. The LFK (Leucine-Phenylalanine-Lysine) motif in nsp14 is completely conserved among the seven known human pathogenic coronaviruses, including SARS-CoV, MERS-CoV and the four endemic strains (Supplementary Fig. 1a). The coexistence of highly conserved metal-binding and LYR-like motifs can be predictive of Fe-S cluster ligation. To explore this possibility, we conducted co-immunoprecipitation (co-IP) and mass spectrometry analyses in HEK293 cells. These experiments revealed that nsp14 transiently interacts with components of both the de novo Fe-S cluster biogenesis system (HSC20, HSPA9, the cysteine desulfurase NFS1, and the main scaffold ISCU) and the cytoplasmic Fe-S assembly (CIA) machinery (CIAO1, MMS19, and FAM96B) (Fig. 1a, b, and Supplementary Data 1). Nsp14 also interacts with PCBP1 and BOLA2 (Fig. 1a, b, and Supplementary Data 1), two proteins that constitute an iron chaperoning system involved in cytoplasmic Fe-S cluster assembly[53]. The interplay among SARS-CoV-2 nsps represents a complex network of interactions central to viral replication. For example, the exoribonuclease activity of nsp14 is activated through its binding to nsp10, which also serves as an accessory factor for nsp16. The structure of nsp10 is notable for its highly conserved metal-binding sites, consisting of Cys$_4$ and Cys$_3$His ligands (Supplementary Fig. 1b). We found that nsp10 interacts with known components of the Fe-S cluster assembly machinery, as well as with PCBP1 and BOLA2 (Fig. 1c, d). Notably, co-expression of nsp10 and nsp14 yielded more co-immunoprecipitated Fe-S biogenesis factors than either nsp alone, consistent with the independent recruitment of the Fe-S assembly machinery by each of the two proteins (Fig. 1c). Formation of these complexes is likely important for Fe-S cluster acquisition by nsp14 and nsp10. In contrast, we observed no such interactions with nsp16 alone (Fig. 1d), in agreement with the absence of any discernible metal-binding sites in nsp16 that could coordinate a Fe-S cluster. To determine whether nsp14 and nsp10 can bind one or more Fe-S clusters, we quantified $^{55}$Fe incorporation into the proteins expressed in cells transfected with either a pool of nontargeting small interfering RNAs (NT si-RNAs) or with si-RNAs targeting the cysteine desulfurase *NFS1* (si-NFS1) (Fig. 1e−g). In control cells (NT si-RNAs), both nsp14 and nsp10 bound radiolabeled iron (10275 ± 1539 and 10225 ± 1212 cpm/mg of cytosolic proteins, respectively), while nsp16 showed no detectable binding (iron levels were comparable to background, corresponding to iron stochastically bound to the beads) (Fig. 1e−g). In cells in which *NFS1* had been silenced (si-NFS1), nsp14 and nsp10 failed to incorporate iron (Fig. 1e, f). These results demonstrate that nsp14 and nsp10 bind iron, likely in the form of Fe-S clusters.

### Spectroscopic and metal content analyses of SARS-CoV-2 nsp14 and nsp10 support ligation of [Fe$_4$S$_4$] clusters

Nsp14 and nsp10, expressed in Expi293F mammalian cells and purified anoxically, displayed a prominent absorbance peak at ~420 nm in their UV-vis absorption spectra (Fig. 2a, b), which is characteristic of Fe-S proteins[54,55]. Notably, co-expression and purification of the two proteins resulted in greater absorption intensity than when the proteins were expressed individually, further supporting the presence of Fe-S clusters in both nsps (Fig. 2a, b). Additionally, nsp14 purified from either mammalian cells (Fig. 2b) or *E. coli* (Fig. 2c, via a pET-28a vector co-expressed with the isc operon from *Azotobacter vinelandii* in pDB1282[56] encoding the Fe-S biogenesis components) exhibited a distinct brown coloration typical of Fe-S proteins (Fig. 2d). However, nsp14 expression in *E. coli* yielded significantly less protein (Fig. 2c) than expression in mammalian cells (Fig. 2b). The continuous-wave electron paramagnetic resonance (CW-EPR) spectrum of nsp14 or nsp10 purified from Expi293F cells, recorded at 10 K, showed no signal (Supplementary Fig. 2a), ruling out the presence of a detectable

quantity of any Fe-S cluster with a half-integer-spin ground state. However, after treating the samples with the reductant dithionite, we observed an EPR signature consistent with a reduced $[Fe_4S_4]^+$ cluster, with $g$ values of 2.05, 1.92, and 1.86 for nsp14, and 2.05, 1.90, and 1.86 for nsp10 (Fig. 2e)[57]. To determine the Fe-S cluster content in nsp14 and nsp10, we performed inductively coupled plasma mass spectrometry (ICP-MS) on proteins expressed and purified anoxically from Expi293F

cells. The analysis included a catalytically inactive nsp14 (nsp14 E191A[22]), a variant with the LFK motif replaced by Ala$_3$, and constructs encoding multiple variants each having one of the metal-binding sites eliminated by substitution of three of its coordinating ligands with serine residues. By this series of Cys/His into serine substitutions, we aimed to identify the Fe-S cluster ligands in nsp10 and nsp14. Wild-type nsp14 bound $7.2 \pm 0.2$ iron ions and $0.9 \pm 0.2$ zinc ions per protomer

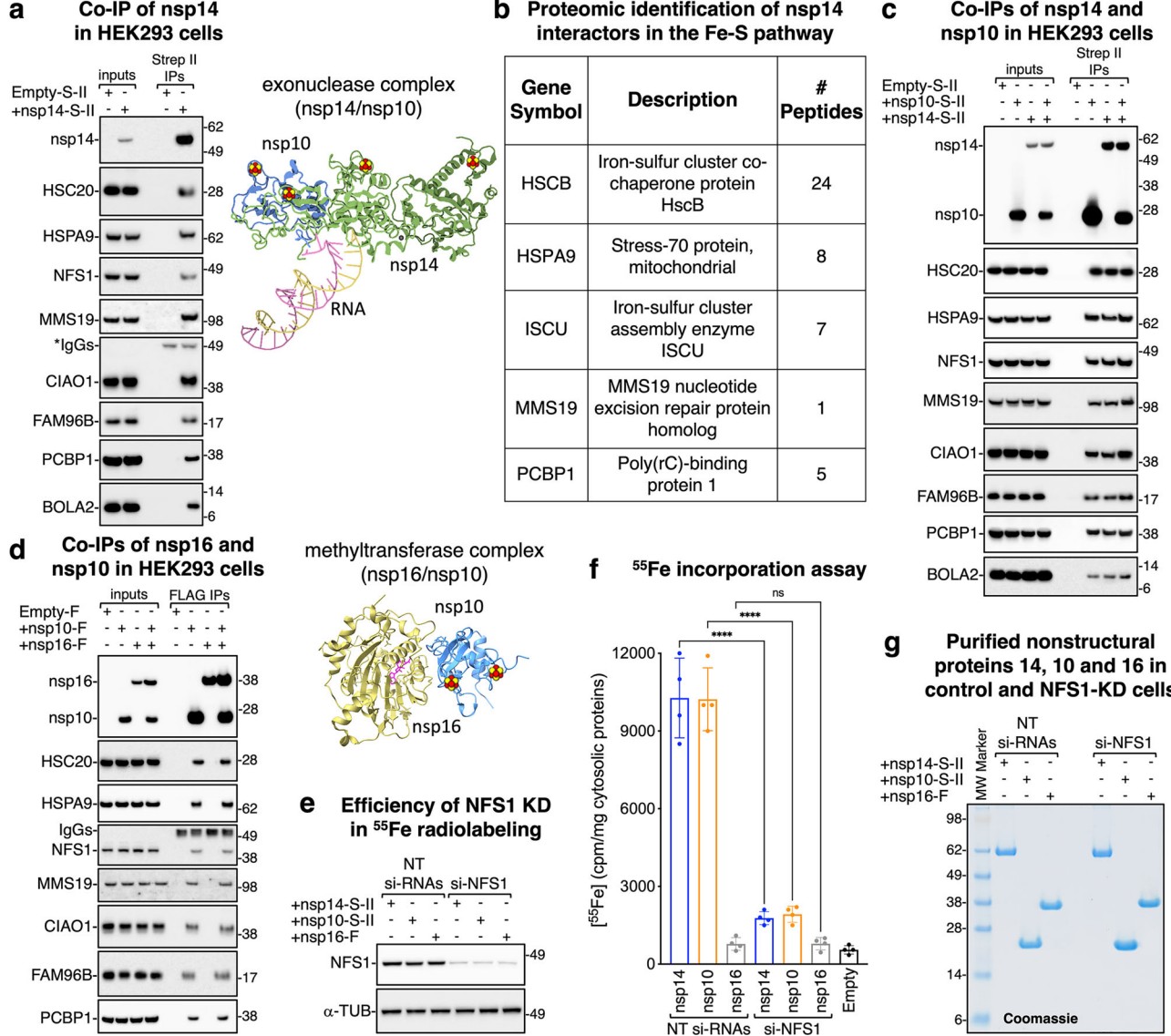

**Fig. 1 | Fe-S cluster incorporation into nsp14 and nsp10 occurs through their interactions with components of the Fe-S biogenesis machinery. a** Co-IP of nsp14 in HEK293 cells. Eluates were probed with antibodies against Strep II and against components of the Fe-S cluster assembly pathway ($n = 4$ independent experiments). The band labeled as IgGs corresponds to immunoglobulin heavy chains. On the right, a cartoon representation of the nsp14/nsp10 complex (PDB ID: 7N0C)[22], is shown, highlighting the metal-binding sites identified as Fe-S cluster ligating centers in the present study. **b** Mass spectrometry identification of affinity-purified interacting partners of nsp14 involved in the Fe-S cluster assembly pathway (see Supplementary Data 1 for a complete list) ($n = 4$ independent experiments). **c** Co-IPs of nsp14 and nsp10 in HEK293 cells ($n = 4$ independent experiments). **d** Co-IPs of nsp16 and nsp10 in HEK293 cells ($n = 4$ independent experiments). On the right, a cartoon representation of the nsp16/nsp10 methyltransferase complex (PDB ID: 6W4H)[79] is shown, highlighting the metal-binding sites in nsp10 identified as Fe-S cluster ligating centers in the present study. **e** Representative immunoblots to NFS1 and to α-tubulin, used as a loading control, in control and *NFS1*-depleted

cells that were quantified in (**f**) for their iron content ($n = 4$ independent experiments). **f** Levels of radioactive iron ($^{55}$Fe) incorporated into nsp14, nsp10 and nsp16 in control cells transfected with non-targeting siRNAs (NT-siRNAs) and in cells transfected with siRNAs directed against the cysteine desulfurase *NFS1* (si-NFS1). Levels of iron stochastically associated with the beads in lysates from cells transfected with a backbone vector (labeled as Empty) are also reported (accounting for $550 \pm 163.3$ cpm/mg of cytosolic proteins) and were not subtracted from measurements of radiolabeled iron incorporated into nsp14, nsp10 and nsp16 in the chart ($n = 4$ independent experiments). Data are presented as mean values $\pm$ SD. Significance was determined by two-way analysis of variance (ANOVA) and Sidak's multiple comparisons test. ****$P < 0.0001$; ns $P > 0.9999$. **g** Representative Coomassie staining showing levels of nsp14, nsp10 and nsp16 in control and NFS1-depleted cells that were quantified in (**f**) for their iron content ($n = 4$ independent experiments). Source data are provided as a Source Data file and in the Supplementary Information.

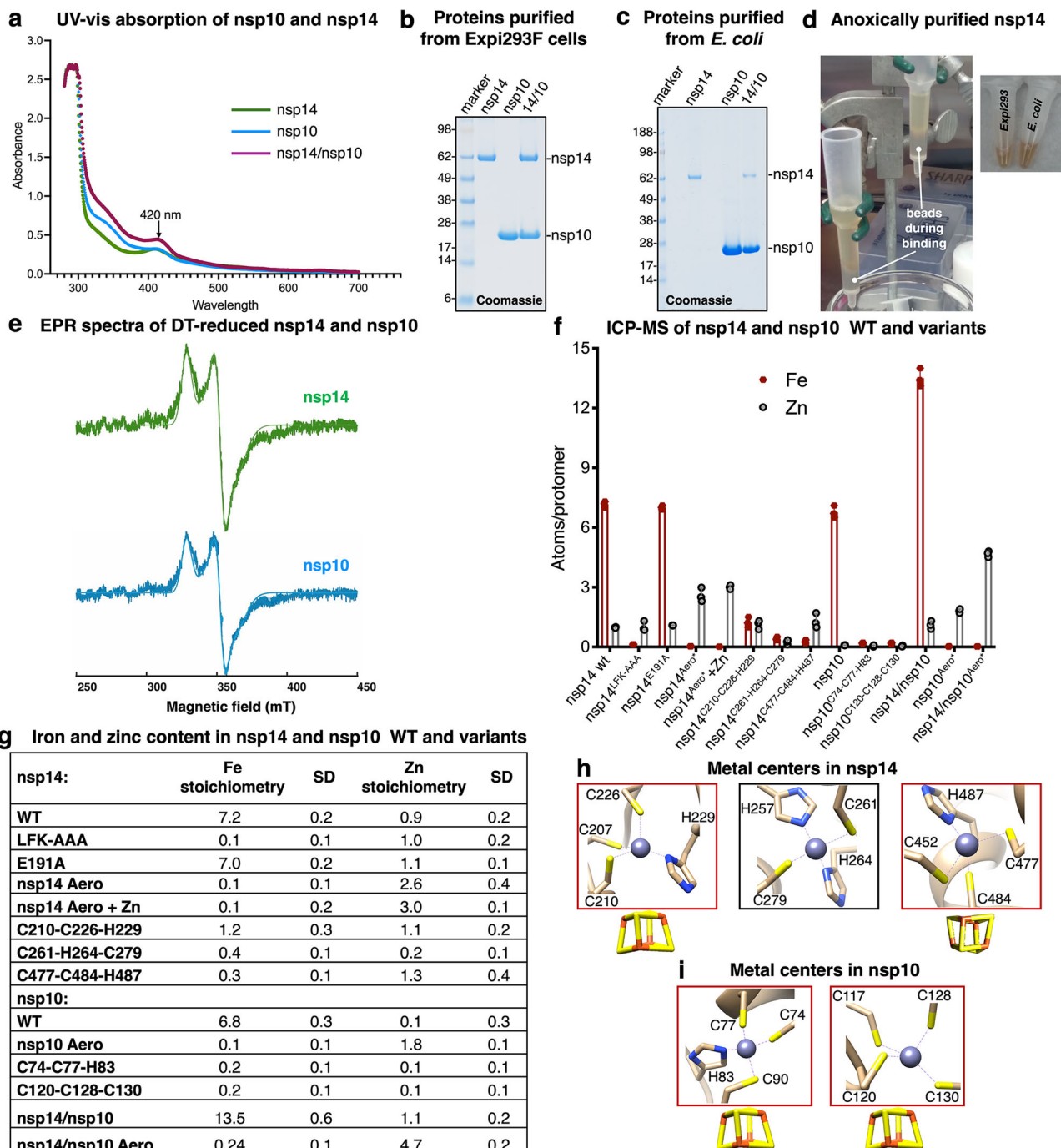

**Fig. 2 | Evidence for ligation of Fe-S metal cofactors by nsp14 and nsp10 expressed in Expi293F cells in previously annotated zinc-binding sites. a** UV-visible absorption spectra of nsp10 and nsp14, individually or co-expressed and purified from Expi293F cells, showing peaks at ~420 nm characteristic of [Fe$_4$S$_4$] clusters. **b** Representative Coomassie staining of purified nsp14 and nsp10, individually or co-expressed and purified from Expi293F cells ($n = 4$ independent experiments). **c** Representative Coomassie staining of purified nsp14 and nsp10, individually or co-expressed in *E. coli* ($n = 3$ independent experiments). **d** Images of columns during the purification of nsp14 from Expi293F and *E. coli* cells are shown on the left. Eluates of purified nsp14 proteins, exhibiting the typical brown color characteristic of Fe-S enzymes, are presented on the right. **e** Continuous-wave EPR (CW-EPR) spectrum recorded at 10 K for nsp14 and nsp10 upon reduction with sodium dithionite (DT). The parameters obtained by simulating the spectrum of the reduced clusters in nsp14 are (**g**) = (2.05, 1.92, 1.86) and for nsp10 are (**g**) = (2.05, 1.90, 1.86). **f** Inductively coupled plasma mass spectrometry (ICP-MS) performed on nsp14 and nsp10 wild-type (WT) and variants, as indicated ($n = 3$ independent experiments). Nsp14$^{Aero*}$ represents nsp14 purified under standard oxygen tension with (+Zn) or without zinc reconstitution. Nsp10$^{Aero*}$ represents nsp10 purified under standard oxygen tension (on the bench) without zinc reconstitution. Nsp14/nsp10 Aero represents the heterodimeric complex of the two proteins co-expressed and purified from mammalian cells without zinc reconstitution. Data are presented as mean values ± SD. **g** Table summarizing the iron and zinc content of nsp14 and nsp10 WT and variant proteins, as determined by ICP-MS. The standard deviation (SD) was calculated from three independent experiments. **h** Metal centers coordinating zinc in the available structure of nsp14 (PDB ID: 7N0C). Two sites identified as Fe-S cluster ligating centers (Cys$_3$His) are highlighted, with a cubane cluster depicted beneath the box for illustrative purposes. **i** Metal centers coordinating zinc in the available structure of nsp10 (PDB ID: 7N0C). Both sites were identified as Fe-S centers in the present study. Source data are provided as a Source Data file and in the Supplementary Information.

(Fig. 2f, g, and Supplementary Fig. 2b). The E191A variant exhibited a similar metal composition, with $7.0 \pm 0.2$ iron ions and $1.1 \pm 0.1$ zinc ions per protomer. In contrast, the LFK motif variant (LFK-AAA) and nsp14 purified under standard aerobic conditions (on the bench, nsp14$^{Aero*}$) lacked iron but retained one and $2.6 \pm 0.35$ zinc ions per protomer, respectively (Fig. 2f, g). Both findings, along with the EPR results, demonstrate that nsp14 ligates two [Fe$_4$S$_4$] clusters. Aerobically purified nsp14, upon zinc reconstitution (nsp14$^{Aero*}$ + Zn), bound 3 zinc ions per protomer (Fig. 2f, g and Supplementary Fig. 2b). Substituting three residues with serines in the first (C207-C210-C226-H229; C210S-C226S-H229S) or third (C452-C477-C484-H487; C477S-C484S-H487S) Cys$_3$His-type metal-binding site abolished iron binding without significantly impacting zinc binding (Fig. 2f, g). In contrast, disruption of the second Cys$_2$His$_2$-type metal-binding site (H257-C261-H264-C279; C261S-H264S-C279S) eliminated binding of both iron and zinc (Fig. 2f, g), likely due to protein misfolding. Determination of the iron content of nsp14 wild-type and variants by using the colorimetric iron indicator, ferrozine and quantitative amino acid analysis confirmed the ICP-MS results (Supplementary Fig. 2c). Iron concentrations in nsp14 WT, nsp14$^{LFK-AAA}$, nsp14$^{E191A}$, nsp14$^{C210S-C226S-H229S}$, nsp14$^{C261S-H264S-C279S}$, nsp14$^{C477S-C484S-H487S}$ each at $50\,\mu M$ were $375 \pm 55\,\mu M$, $15 \pm 3\,\mu M$, $361 \pm 52\,\mu M$, $51 \pm 13\,\mu M$, $25 \pm 11\,\mu M$, and $65 \pm 14\,\mu M$ respectively, corresponding to $7.5 \pm 1.9$ iron/protomer for nsp14 WT, $0.3 \pm 0.1$ for nsp14$^{LFK-AAA}$, $7.2 \pm 1.0$ for nsp14$^{E191A}$, $1.0 \pm 0.2$ for nsp14$^{C210S-C226S-H229S}$, $0.2 \pm 0.2$ for nsp14$^{C261S-H264S-C279S}$, and $1.0 \pm 0.3$ for nsp14$^{C477S-C484S-H487S}$ (Supplementary Fig. 2c). These findings demonstrate the presence of one [Fe$_4$S$_4$] cluster at each Cys$_3$His site and one zinc ion at the Cys$_2$His$_2$ center in nsp14 overexpressed in mammalian cells.

By ICP-MS analysis of wild-type nsp10, we found that the protein bound $6.8 \pm 0.3$ iron ions per protomer (Fig. 2f, g and Supplementary Fig. 2b). Nsp10 purified under standard aerobic conditions (on the bench, nsp10$^{Aero*}$) lacked iron but retained $1.8 \pm 0.1$ zinc ions per protomer (Fig. 2f, g). Disruption of either metal-binding center in nsp10 by replacing three ligands of the Cys$_3$His (C74-C77-H83-C90) or Cys$_4$ (C117-C120-C128-C130) center with serines eliminated iron binding, likely due to protein misfolding, indicating that nsp10 binds two Fe-S clusters, likely of the cubane Fe$_4$S$_4$ type. Determination of the iron content of nsp10 wild-type and variants via ferrozine assay and quantitative amino acid analysis confirmed the ICP-MS results (Supplementary Fig. 2c). Iron concentrations for nsp10 WT, nsp10$^{C74S-C77S-H83S}$ and nsp10$^{C120S-C128S-C130S}$ each at $50\,\mu M$ were $340 \pm 27\,\mu M$, $23 \pm 21\,\mu M$ and $35 \pm 18\,\mu M$, respectively, corresponding to $6.8 \pm 0.5$ iron/protomer for nsp10 WT, $0.5 \pm 0.4$ for nsp10$^{C74S-C77S-H83S}$ and $0.7 \pm 0.4$ for nsp10$^{C120S-C128S-C130S}$ (Supplementary Fig. 2c). Overall, EPR spectroscopy and iron stoichiometry demonstrate the presence of two [Fe$_4$S$_4$] clusters in nsp10 expressed and purified from mammalian cells.

Nsp14 and nsp10 coexpressed and purified from mammalian cells bound $13.5 \pm 0.6$ iron ions and $1.1 \pm 0.2$ zinc ions (Fig. 2f, g). Overall, these results confirm the presence of [Fe$_4$S$_4$] clusters at the Cys$_3$His sites and one zinc ion in the Cys$_2$His$_2$ site of nsp14 (Fig. 2h), and of two [Fe$_4$S$_4$] clusters in nsp10 (Fig. 2i). The suboptimal iron occupancy determined in these measurements likely reflects the presence of a pool of apo-proteins in the purification batches.

## Fe-S clusters in nsp14 and nsp10 enhance the methyltransferase activities of nsp14 and the nsp16/nsp10 complex

We next aimed to characterize the role of the clusters in nsp14 and nsp10 by performing functional assays of the exoribonuclease (ExoN) activity of the nsp10/nsp14 complex and the methyltransferase activities of nsp14 and the nsp16/nsp10 complex.

To assay the ExoN activity of the nsp10/nsp14 complex (Supplementary Fig. 3a), we employed an RNA substrate consisting of a template strand with three initiating cytidines followed by 31 nucleotides (nts) (Fig. 3a). This strand also served as the nonscissile strand for ExoN. The complementary product strand, which served as the scissile strand for ExoN, was 29 nts in length and terminated with a cytidine-5'-monophosphate (CMP), creating a C-U mismatch at the 3'-end (Fig. 3a). The scissile strand was modified at the 5'-end to incorporate a fluorophore, Alexa488, a fluorescein-based dye derivative known for its photostability, strong fluorescence signal, and compatibility with RNA substrates. Its placement ensures that fluorescence changes can be linked to the progression of the exonuclease activity.

The ExoN activity assay of the nsp14/nsp10 complex was monitored from 0 min (t0) to 30 min (t30). The assay included wild-type nsp14 (nsp14 WT) harboring either two [Fe$_4$S$_4$] clusters (Fe-S nsp14) or zinc at all three metal-binding sites (All Zn), in complex with nsp10, coordinating Fe-S clusters at its two metal-binding centers. Catalytically inactive nsp14 (E191A) and a variant in which the LFK motif was replaced by triple alanines were also tested (Fig. 3a and Supplementary Fig. 3b, top panels).

The nsp14/nsp10 complex was obtained by co-expressing the two proteins (wild-type or variants, as indicated) in Expi293F mammalian cells and copurifying them under anoxic conditions, except for the complexes labeled "All Zn", which were co-expressed and purified on the bench under aerobic conditions and contained only zinc, as confirmed by ICP-MS analysis (Fig. 2f, g).

Both Fe-S-nsp14 and zinc-containing nsp14 (All Zn), when complexed with nsp10 containing its native Fe-S clusters, efficiently digested the RNA substrate (Fig. 3a and Supplementary Fig. 3b, top panels and lower panels, left side). In contrast, neither the E191A nor the LFK variant exhibited ExoN activity. Testing Zn-nsp10 in complex with either Fe-S-nsp14 or zinc-nsp14 revealed that both enzymes were equally efficient in cleaving the RNA substrate (Fig. 3a and Supplementary Fig. 3b, bottom panels, right side). These results demonstrate that the ExoN activity of the nsp14/nsp10 complex is unaffected by which metal species is bound.

Nsp14 and the nsp16/nsp10 complex methylate Gppp-RNA at two specific positions: N7 of the guanine[14,15] and 2'-O of the first nucleotide ribose[15,58], resulting in the formation of the Cap-0 and Cap-1 structures, respectively (Fig. 3b). To evaluate the methyltransferase (MTase) activities of these enzymes, we used a bioluminescence-based assay that monitors the conversion of S-adenosyl-methionine (SAM) into S-adenosyl homocysteine (SAH), the reaction byproduct. SAH is then converted into ADP and subsequently into ATP using the MTase-Glo™ Assay, which, via a luciferase reaction, generates a luminescent signal proportional to the amount of ATP generated from SAH in the reaction mix (Fig. 3c). The luminescence signal allows for quantitation of SAH by comparison to a standard curve (Supplementary Fig. 4a).

Because nsp14 functions primarily as an RNA MTase on uncapped 5'-guanosine triphosphate substrates, we assessed its activity with both uncapped RNA (RNA-1) and capped RNA (Cap-0, methylated at the N7 position; Fig. 3c) to determine its substrate specificity and catalytic efficiency under the tested conditions. Two forms of nsp14 were evaluated: one harboring two [Fe$_4$S$_4$] clusters (WT-Fe-S-nsp14) and the other purified aerobically and containing only zinc (WT-Zn-nsp14). Cap-0 proved to be a poor substrate for both enzyme forms (Fig. 3c). Despite optimizing assay conditions, including increased enzyme concentrations and extended reaction times, the signal remained very low and comparable to the inactive LFK variant (Supplementary Fig. 4b, 4c). Although the LFK substitution resides in the N-terminal domain of nsp14 (Supplementary Fig. 1a), it disrupts the incorporation of both Fe-S clusters (Fig. 2f, g), including the one in the C-terminal domain, which supports methyltransferase activity (Fig. 2h), likely leading to misfolding and loss of enzymatic function (Supplementary Fig. 4c). These results are consistent with the established selectivity of nsp14 for uncapped RNA substrates (RNA-1)[14,15]. Notably, relative to Zn-nsp14, Fe-S-nsp14 exhibited: i. greater maximum activity (Bmax; Fig. 3c−e) and ii. 4.5 and 1.6 times higher affinity for its RNA-1 (Fig. 3d) and SAM (Fig. 3e) substrates, respectively. Collectively, these findings highlight a role of Fe-S clusters in enhancing the methyltransferase activity and substrate affinities of nsp14.

## a ExoN activity assay of the nsp14/nsp10 complex

nucleotide sequence of the RNA substrate for ExoN activity

```
3'-GCUUUCCCUUUCCCUUCCUCCUUCUUUCUUUCCC-5'
5'-CGAAAGGGAAAGGGAAGGGGAAGAAAGAC-3'
     /5Alexa488N/
```

secondary structure of the RNA substrate for ExoN activity

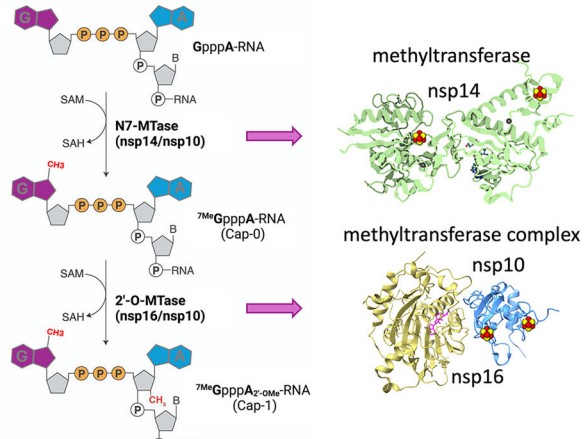

nsp14 exoribonuclease activity

## c Methyltransferase activity assay of nsp14

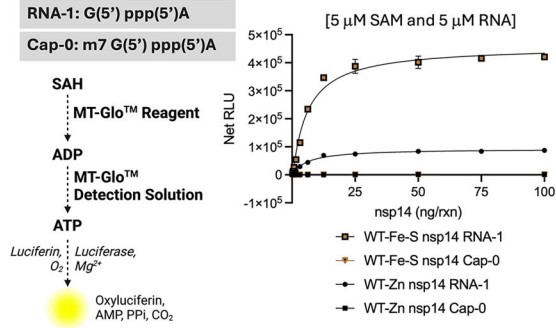

| Specific binding with Hill slope | WT-Zn nsp14 RNA-1 | WT-Fe-S nsp14 RNA-1 |
|---|---|---|
| Best-fit values | | |
| Bmax (95% CI) | 88729 (86489 to 91214) | 420538 (412164 to 429380) |
| h (95% CI) | 1.23 (1.13 to 1.34) | 1.64 (1.50 to 1.80) |
| Kd (95% CI) | 5.61 (5.17 to 6.12) | 5.36 (5.04 to 5.70) |

## d Methyltransferase activity assay of nsp14

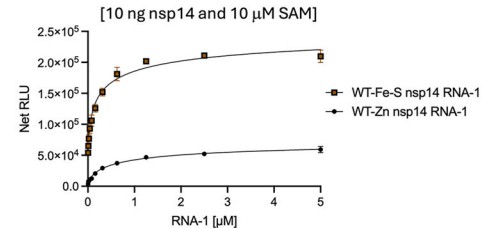

| Specific binding with Hill slope | WT-Zn nsp14 RNA-1 | WT-Fe-S nsp14 RNA-1 |
|---|---|---|
| Best-fit values | | |
| Bmax (95% CI) | 77818 (69001 to 92246) | 270582 (242416 to 319866) |
| h (95% CI) | 0.62 (0.54 to 0.71) | 0.43 (0.36 to 0.51) |
| Kd (95% CI) | 0.74 (0.47 to 1.44) | 0.17 (0.10 to 0.44) |

## b Schematic of the steps required for viral RNA capping

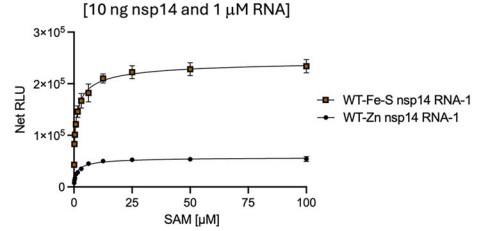

## e Methyltransferase activity assay of nsp14

| Specific binding with Hill slope | WT-Zn nsp14 RNA-1 | WT-Fe-S nsp14 RNA-1 |
|---|---|---|
| Best-fit values | | |
| Bmax (95% CI) | 58408 (55321 to 62320) | 250269 (235225 to 271136) |
| h (95% CI) | 0.70 (0.60 to 0.81) | 0.58 (0.48 to 0.68) |
| Kd (95% CI) | 1.40 (1.10 to 1.87) | 0.87 (0.65 to 1.30) |

**Fig. 3 | Fe-S clusters in nsp14 specifically enhance methyltransferase activity without affecting exoribonuclease (ExoN) function. a** From top to bottom: nucleotide sequence and secondary structure of the RNA substrate used in the ExoN activity assay, modeled with RNafold and RNA Composer. ExoN activity assay of the nsp14/nsp10 complex monitored from time 0 (t0) to 30 min (t30). The assay was performed with nsp14 WT ligating either two Fe-S clusters (Fe-S nsp14) at the Cys₃His site or zinc at all three metal-binding sites (All Zn), in complex with nsp10 coordinating Fe-S clusters at its two metal-binding centers. Nsp14, either ligating two Fe-S clusters or zinc, was also tested in the presence of nsp10 ligating zinc (Zn-nsp10; lower panel) (*n* = 3 independent experiments). **b** Schematic of the sequential steps required for viral RNA capping (Created in BioRender. Maio, N. (2025) https://BioRender.com/1j8x1bi). **c** Bioluminescence-based methyltransferase activity assay of nsp14 conducted with increasing concentrations of nsp14, as indicated, over 30 min in the presence of 5 μM S-adenosylmethionine (SAM) and 5 μM RNA. Data are presented as mean values ± SD (*n* = 3 independent experiments). A schematic is included illustrating the conversion of S-adenosylhomocysteine (SAH) into ATP in the assay. The kinetic parameters shown in the table were determined from three independent experiments, with standard deviations (SD) shown as error bars. **d** Bioluminescence-based methyltransferase assay of nsp14 conducted with increasing concentrations of the substrate (RNA-1) over 30 min in the presence of 10 ng nsp14 and 10 μM SAM. Data are presented as mean values ± SD (*n* = 3 independent experiments). The kinetic parameters shown in the table were determined from three independent experiments, with standard deviations (SD) shown as error bars. **e** Bioluminescence-based methyltransferase assay of nsp14 conducted with increasing concentrations of SAM over 30 min in the presence of 10 ng nsp14 and 1 μM RNA. Data are presented as mean values ± SD (*n* = 3 independent experiments). The kinetic parameters shown in the table were determined from three independent experiments, with standard deviations (SD) shown as error bars. Source data are provided as a Source Data file and in the Supplementary Information.

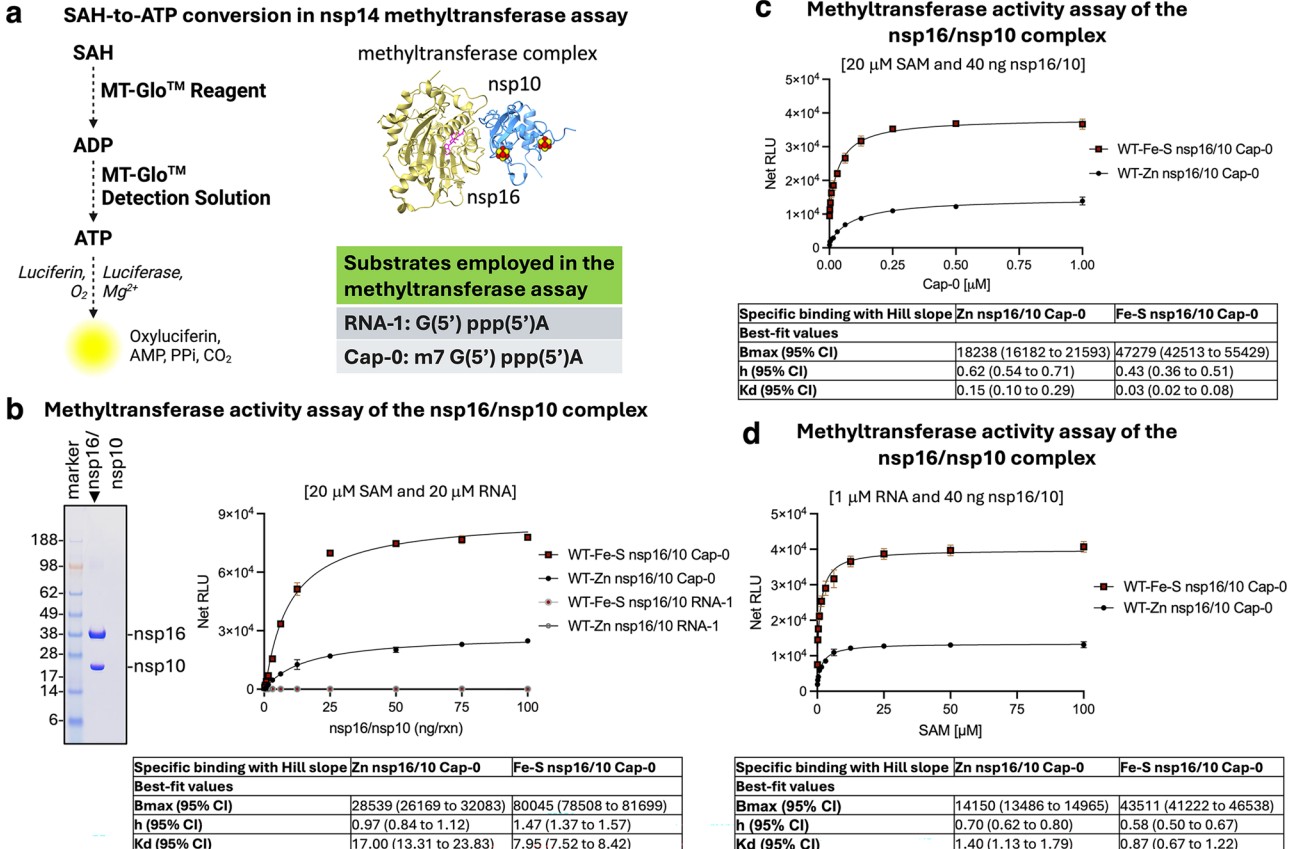

**a** SAH-to-ATP conversion in nsp14 methyltransferase assay

**b** Methyltransferase activity assay of the nsp16/nsp10 complex

| Specific binding with Hill slope | Zn nsp16/10 Cap-0 | Fe-S nsp16/10 Cap-0 |
|---|---|---|
| **Best-fit values** | | |
| Bmax (95% CI) | 28539 (26169 to 32083) | 80045 (78508 to 81699) |
| h (95% CI) | 0.97 (0.84 to 1.12) | 1.47 (1.37 to 1.57) |
| Kd (95% CI) | 17.00 (13.31 to 23.83) | 7.95 (7.52 to 8.42) |

**c** Methyltransferase activity assay of the nsp16/nsp10 complex

| Specific binding with Hill slope | Zn nsp16/10 Cap-0 | Fe-S nsp16/10 Cap-0 |
|---|---|---|
| **Best-fit values** | | |
| Bmax (95% CI) | 18238 (16182 to 21593) | 47279 (42513 to 55429) |
| h (95% CI) | 0.62 (0.54 to 0.71) | 0.43 (0.36 to 0.51) |
| Kd (95% CI) | 0.15 (0.10 to 0.29) | 0.03 (0.02 to 0.08) |

**d** Methyltransferase activity assay of the nsp16/nsp10 complex

| Specific binding with Hill slope | Zn nsp16/10 Cap-0 | Fe-S nsp16/10 Cap-0 |
|---|---|---|
| **Best-fit values** | | |
| Bmax (95% CI) | 14150 (13486 to 14965) | 43511 (41222 to 46538) |
| h (95% CI) | 0.70 (0.62 to 0.80) | 0.58 (0.50 to 0.67) |
| Kd (95% CI) | 1.40 (1.13 to 1.79) | 0.87 (0.67 to 1.22) |

**Fig. 4 | Putative zinc/ metal-binding sites in nsp10 ligate two Fe-S clusters that enhance the methyltransferase activity of the nsp16/nsp10 complex.**
**a** Schematic illustrating the conversion of SAH to ATP in the methyltransferase assay. Structural cartoon of the methyltransferase nsp16/nsp10 complex (PDB ID: 6W4H), highlighting the Fe-S clusters in nsp10. The substrates used, RNA-1 (G(5′) ppp(5′)A) and Cap-0 (m7G(5′)ppp(5′)A), are indicated. **b** Representative Coomassie staining of nsp16/nsp10 co-expressed and purified from Expi293F mammalian cells (left). Bioluminescence-based methyltransferase activity assay of the nsp16/nsp10 complex performed with increasing concentrations of nsp16/nsp10 over 30 min, in the presence of 20 µM SAM and 20 µM RNA. Data are presented as mean values ± SD ($n = 3$ independent experiments). Table summarizing the kinetic parameters of reactions involving the nsp16/nsp10 complex, with nsp10 coordinating either zinc (WT-Zn nsp16/nsp10) or two Fe-S clusters (WT-Fe-S nsp16/nsp10), in the presence of Cap-0. Kinetic parameters were determined from three independent experiments. **c** Bioluminescence-based methyltransferase activity assay of the

nsp16/nsp10 complex performed with increasing concentrations of Cap-0 over 30 min, in the presence of 20 µM SAM and 40 ng nsp16/nsp10. Data are presented as mean values ± SD ($n = 3$ independent experiments). Table summarizing the kinetic parameters of reactions involving the nsp16/nsp10 complex, with nsp10 coordinating either zinc (WT-Zn nsp16/nsp10) or two Fe-S clusters (WT-Fe-S nsp16/nsp10), in the presence of Cap-0. Kinetic parameters were determined from three independent experiments. **d.** Bioluminescence-based methyltransferase activity assay of the nsp16/nsp10 complex performed with increasing concentrations of SAM over 30 min, in the presence of 1 µM RNA (Cap-0) and 40 ng nsp16/nsp10. Data are presented as mean values ± SD ($n = 3$ independent experiments). Table summarizing the kinetic parameters of reactions involving the nsp16/nsp10 complex, with nsp10 coordinating either zinc (WT-Zn nsp16/nsp10) or two Fe-S clusters (WT-Fe-S nsp16/nsp10), in the presence of Cap-0. Kinetic parameters were determined from three independent experiments. Source data are provided as a Source Data file and in the Supplementary Information.

We applied the same assay to the nsp16/nsp10 complex (Fig. 4a) and found that the complex with nsp10 ligating two Fe-S clusters (WT-Fe-S nsp16/nsp10) exhibited a 3-fold greater Bmax than that with the zinc-bound form (WT-Zn nsp16/nsp10; Fig. 4b). Additionally, WT-Fe-S nsp16/nsp10 showed a 5-fold greater affinity for its physiological substrate, Cap-0 (Fig. 4c), and a 1.6-fold greater affinity for SAM (Fig. 4d) than the zinc enzyme. Both enzyme forms displayed weak activity on RNA-1, consistent with their preference for N7-methylated RNA substrates. These results demonstrate the enhanced catalytic efficiency and substrate affinity conferred by Fe-S clusters in the nsp16/nsp10 complex.

**Comparative reduction potentials of Fe-S clusters in SARS-CoV-2 methyltransferase capping enzymes: more electronegative in nsp10 than in nsp14**

Overall, our findings highlight the functional significance of Fe-S clusters in the methyltransferase activities of SARS-CoV-2 capping enzymes. To gain insights into the properties of these cofactors, we

calculated their reduction potentials for the $[Fe_4S_4]^{2+/1+}$ couple, following the QM/MM approach employed in our prior work on Fe-S ligating peptides[59], to which we refer for additional details. The reduction potentials for the $Cys_3His$- and $Cys_4$-ligated cofactors of nsp10 were -0.19 and -0.26 V, respectively (Fig. 5a), in agreement with the lower reduction potential of $Cys_4$-coordinated clusters compared to $Cys_3His$[60]. Accordingly, a similar trend was observed in the [FeFe]-hydrogenase from *Clostridium acetobutylicum*[61], where a Cys-to-His substitution resulted in a + 0.07 V shift. Both reduction potentials in nsp10 are lower than those of the $Cys_3His$ coordinated clusters in nsp14, which we found to be -0.04 and -0.07 V (Fig. 5b). We notice that all calculated values fall within the typical range for $Cys_3His$- and $Cys_4$-coordinated Fe-S clusters[60], supporting the validity of our QM/MM approach in capturing key electronic and structural properties of the cofactors and their environments. Our results suggest that the clusters in nsp14 have higher electron affinities than those of nsp10, which may provide a framework for understanding the potential electron transfer dynamics within the RTC (Fig. 5c).

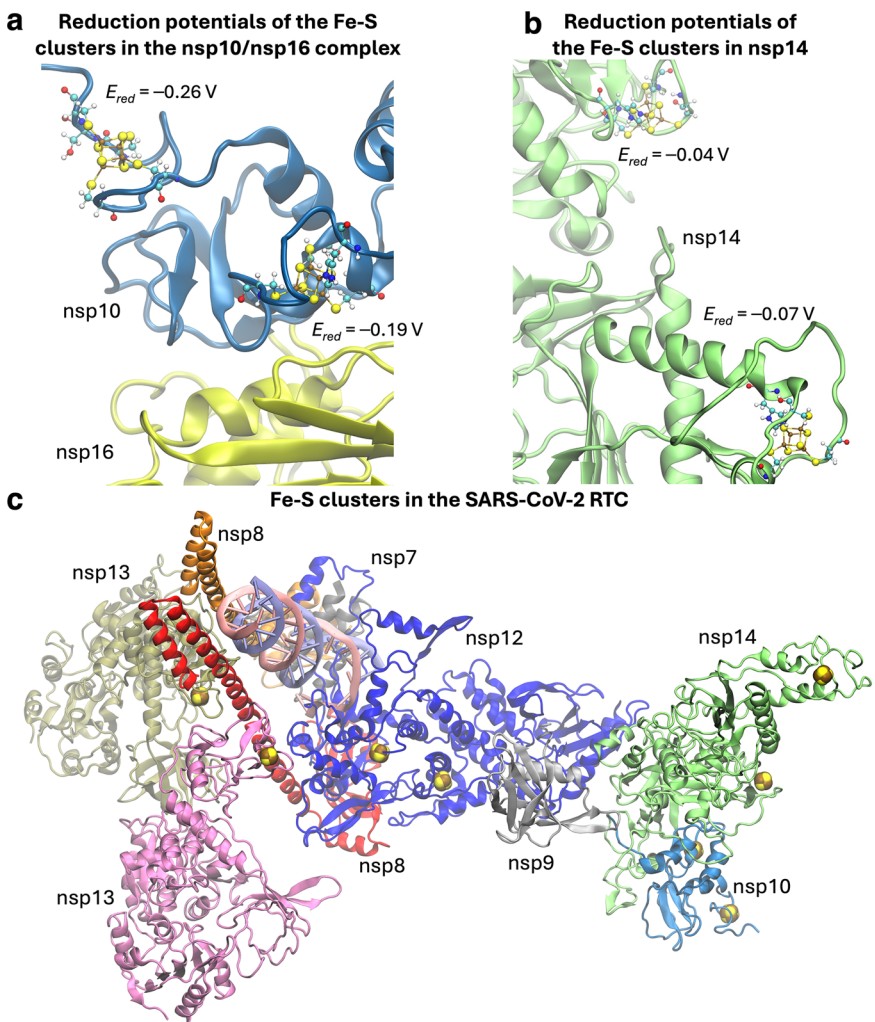

**a** Reduction potentials of the Fe-S clusters in the nsp10/nsp16 complex

$E_{red}$ = −0.26 V

$E_{red}$ = −0.19 V

nsp10

nsp16

**b** Reduction potentials of the Fe-S clusters in nsp14

$E_{red}$ = −0.04 V

nsp14

$E_{red}$ = −0.07 V

**c** Fe-S clusters in the SARS-CoV-2 RTC

nsp8
nsp13
nsp7
nsp12
nsp14
nsp8
nsp9
nsp10
nsp13

**Fig. 5 | Reduction potentials of Fe-S clusters in nsp10 and nsp14 as calculated by density functional theory and structural modeling. a, b** Zoomed-in views of the nsp16/nsp10 (PDB ID: 6W4H[79]) and nsp14 (PDB ID: 7QGI[80]) in cartoon representation. Fe-S clusters and their coordinating ligands are displayed in the CPK style, together with their corresponding reduction potentials. **c** Modeling of [Fe$_4$S$_4$] clusters in SARS-CoV-2 RTC components that were characterized as Fe-S proteins. The complex includes nsp12, nsp8, nsp7, nsp9, nsp10, nsp13, and nsp14 in a 1:2:1:1:1:2:1 stoichiometric ratio (PDB ID: 7EIZ[31]). In the model, nsp12 is represented in blue, nsp8 protomers in red and orange, nsp7 in gray, nsp9 in dark gray, nsp10 in cornflower blue, nsp13 protomers in olive green and pink, and nsp14 in green.

## The Fe-S cluster redox state modulates the RNA binding activities of nsp14 and nsp10

To investigate whether the Fe-S clusters in nsp14 and nsp10 are redox-active and how their redox state influences RNA binding, we treated the proteins with chemical reductants (ascorbate, dithionite) and an oxidant (ferricyanide) to shift the cluster valence from the resting [2 + ] state toward reduced [1 + ] or oxidized [3 + ] states, and assessed their binding to RNA-1 (nsp14) or Cap-0 structures (nsp10/nsp16). Ascorbate acts as a mild reductant (reduction potential -0.081 V vs SHE at pH 7[62]), dithionite as a strong reductant (reduction potential -0.66 V[57]), and ferricyanide as a potent oxidant (reduction potential +0.36 V vs SHE at pH 7[63,64]); all three are widely used to modulate Fe-S cluster redox states in vitro. Attempts to oxidize the Fe-S clusters in nsp14 or nsp10 using ferricyanide resulted in cluster degradation, as reflected by the disappearance of the characteristic absorbance peak in UV–visible spectra, indicating that the clusters cannot undergo a stable [2 + /3 + ] redox transition (Supplementary Fig. 5).

We next assessed the reduction of Fe-S clusters in nsp14 and nsp10 using either ascorbate or dithionite. The Fe-S clusters in nsp14 were susceptible to reduction by ascorbate, with a two-fold molar excess resulting in a 40% reduction and a ten-fold excess achieving 77%

reduction (Fig. 6a). In contrast, the clusters in nsp10 were much less efficiently reduced by ascorbate: only 16% reduction was observed with a ten-fold excess, and 32% with a hundred-fold excess (Fig. 6d). This differential reactivity suggests that the clusters in nsp10 are more electronegative, displaying a lower affinity for electrons and thus a reduced propensity to accept electrons from ascorbate. These findings align with the calculated reduction potentials, which indicate lower reduction potentials (i.e., less favorable reduction) for nsp10 clusters compared to those in nsp14 (Fig. 5a, b). Dithionite, a much stronger reductant, was markedly more effective. A two-fold excess of dithionite reduced the clusters in nsp14 by 90%, while a ten-fold excess reduced the clusters in nsp10 by 80%. These results demonstrate that, despite their lower reduction potential, the clusters in nsp10 can be reduced efficiently with a sufficiently strong reductant, consistent with their calculated redox properties and the EPR spectra (Fig. 2e), showing a more intense signal for nsp14 upon reduction, further corroborating the greater reducibility of its clusters compared to nsp10. We next examined whether redox-induced changes in Fe-S cluster oxidation states affected RNA binding by nsp14 and the nsp10/nsp16 complex. A ten-fold excess of ascorbate, which effectively reduced the clusters in nsp14, markedly impaired its binding to RNA-1 (Fig. 6b, c). In

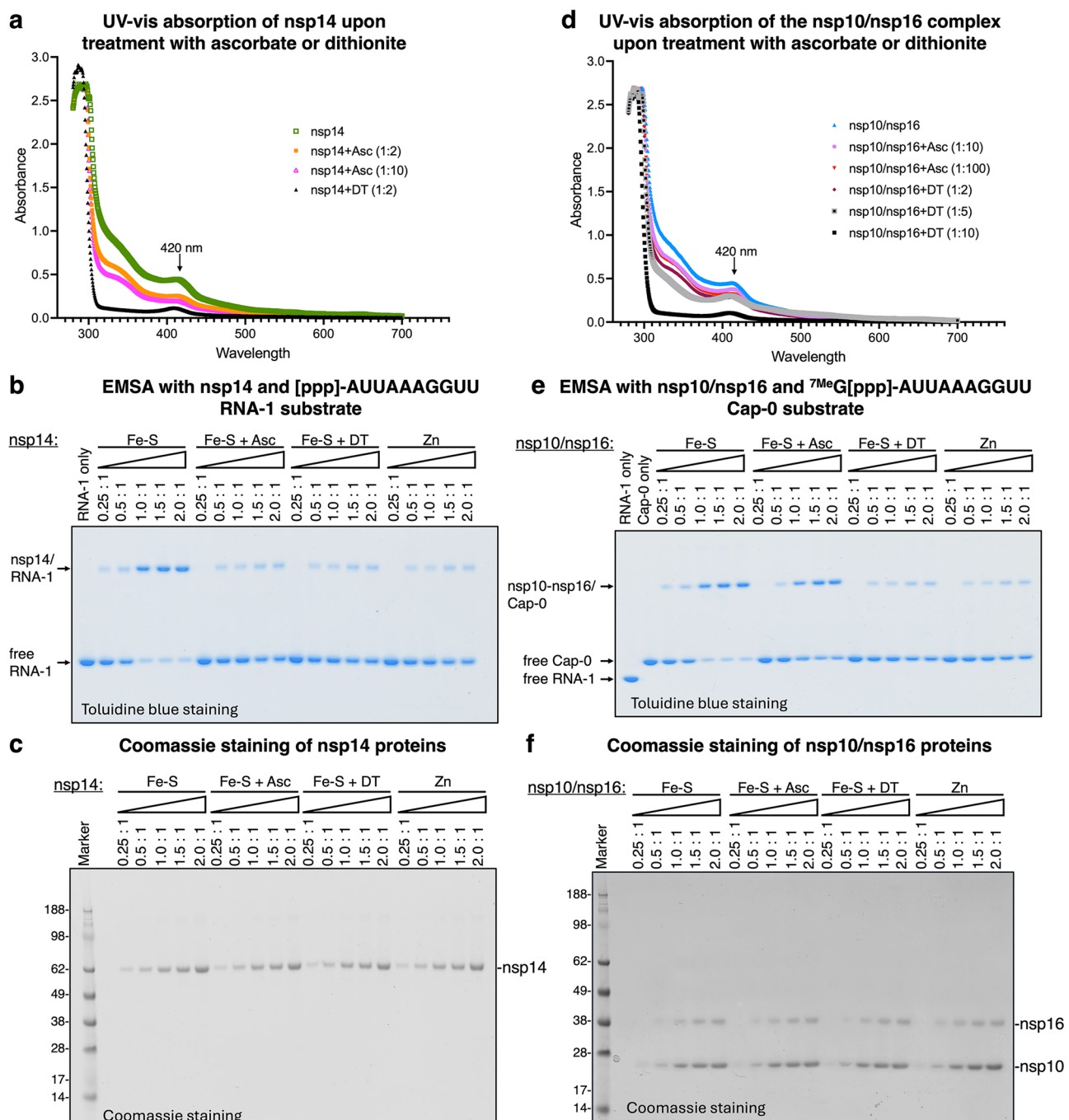

**Fig. 6 | Chemical reduction of Fe-S clusters in nsp14 and nsp10 impairs RNA binding. a** UV-visible absorption spectra of purified nsp14 before and after reduction with ascorbate (2- and 10-fold molar excess) and sodium dithionite (2-fold excess). Both reductants diminish the characteristic Fe-S cluster absorbance, indicating successful reduction. **b** Electrophoretic mobility shift assay (EMSA) showing RNA binding by untreated and reduced nsp14 proteins, alongside the zinc-only nsp14 ligating zinc at all three metal-binding sites. Reduction of Fe-S clusters (10-fold molar excess of either ascorbate or dithionite) markedly diminished RNA binding, similar to the zinc-bound protein ($n = 3$ independent experiments). **c** Coomassie-stained SDS-PAGE of the proteins (1000-fold concentrated) used in panel b, confirming comparable purity and loading ($n = 3$ independent

experiments). **d** UV-visible spectra of the nsp10/nsp16 complex after treatment with ascorbate (10- and 100-fold excess) and dithionite (2-, 5-, and 10-fold excess). Only dithionite at 10-fold excess effectively reduced the Fe-S cluster absorbance signature. **e** EMSA of nsp10/nsp16 complex treated with a ten-fold molar excess of either ascorbate or dithionite, along with the zinc-only nsp10. Reduction with dithionite significantly impaired RNA binding, again resembling the binding behavior of the zinc-containing complex ($n = 3$ independent experiments). **f** Coomassie-stained gel of the proteins (1000-fold concentrated) used in panel (**e**), verifying equivalent protein amounts ($n = 3$ independent experiments). Source data are provided as a Source Data file and in the Supplementary Information.

contrast, the same treatment had no effect on the RNA binding activity of nsp10/nsp16 (Fig. 6e, f), consistent with the limited reduction of nsp10 clusters under these conditions (Fig. 6d). However, treatment with a ten-fold excess of dithionite, which robustly reduced clusters in both proteins, led to a profound loss of RNA-binding activity by both nsp14 and nsp10/nsp16 (Fig. 6b, e). Notably, zinc-substituted forms of nsp14 and nsp10/nsp16 exhibited similarly impaired RNA binding (Fig. 6b and e). Together, these findings suggest that the redox state of

the Fe-S clusters affects the RNA-binding function, and that reduction from the resting [2 +] to the [1 +] state impairs this activity.

## Discussion

Here, we demonstrate that nsp14 and nsp10, critical components of the SARS-CoV-2 replication machinery, when purified anoxically from mammalian cells, harbor Fe-S clusters at metal-binding sites previously assigned as zinc centers. Specifically, we demonstrate the presence of two $[Fe_4S_4]$ clusters and one zinc ion in nsp14 and two $[Fe_4S_4]$ clusters in nsp10. These metal centers enhance the methyltransferase activities of nsp14 and the nsp16/nsp10 complex without affecting the exoribonuclease function of the nsp14/nsp10 complex. The presence of Fe-S clusters in multiple SARS-CoV-2 proteins (Fig. 5c) highlights a reliance of the virus on host biosynthetic pathways for Fe-S incorporation into the replication and transcription complex core components, including the RdRp[23], the helicase nsp13[24], the proofreading exoribonuclease complex nsp14/nsp10, and the RNA capping methyltransferase machinery consisting of nsp14 and the nsp16/nsp10 complex.

Fe-S clusters are among the oldest and most versatile cofactors in nature, dating back to the primordial soup era approximately 3.5 to 4 billion years ago[65]. All coronaviruses are thought to trace back to a common ancestor dating tens of millions of years ago[66]. Thus, the presence of Fe-S clusters in nsp12 (RdRp), nsp13, nsp14, and nsp10 underscores the conservation of ancient cofactors in a virus that emerged more recently in evolutionary history, linking primordial biochemistry to essential mechanisms of viral replication. The Fe-S cluster ligands in the nonstructural proteins are completely conserved among the seven coronaviruses known to infect humans[23,24,67] (Supplementary Fig. 1), highlighting their functional importance and their potential as therapeutic targets against a broad spectrum of coronaviruses, including future emerging strains.

The conservation of these sites suggests that the cofactors must provide some advantages to SARS-CoV-2, likely by enhancing the efficiency and regulation of its replication and transcription processes. But how exactly?

Our findings show that redox changes in the Fe-S clusters of nsp14 and nsp10 markedly influence RNA binding: reduction of the clusters diminishes nsp14 binding to RNA-1 and nsp10/nsp16 binding to Cap-0 structures. These effects suggest a regulatory role for the clusters in RNA engagement rather than direct participation in catalysis. Consistently, the 3′–5′ exoribonuclease activity of the nsp14/nsp10 complex remained unchanged between the zinc-only and Fe-S–bound forms, whereas the methyltransferase activity was moderately increased in the presence of Fe-S clusters, pointing to a model in which the clusters may support methyltransferase function by modulating RNA positioning or local structure. We monitored the redox transitions of the clusters in nsp14 and nsp10 using UV–visible absorption spectroscopy, which reliably reported on oxidation and reduction. Although EPR confirmed redox activity consistent with a $[Fe_4S_4]^{2+}$ to $[Fe_4S_4]^+$ transition upon dithionite reduction, assignment of signals to individual clusters was hindered by the presence of two Fe-S clusters in both nsp14 and nsp10 and the inability of EPR to resolve geometric structures or distinguish electronic signatures of similar clusters. Our attempts to generate single-cluster variants for EPR analysis were unsuccessful, as substitution of ligands in either metal-binding site abolished iron incorporation (Fig. 2f, g), precluding selective spectroscopic interrogation.

We speculate that the presence of multiple Fe-S clusters within distinct components of the RTC reflects an evolutionary strategy to enable redox-based coordination of genome replication and transcription. Analogous to electron transfer chains in nitrogenase[68], complex I of the respiratory chain[69,70], and photosynthetic systems[70,71], these clusters may facilitate conformational coupling through redox transitions. In this model, stepwise changes in the oxidation state of Fe-S clusters, potentially accompanied by ligand rearrangements, could drive coordinated structural shifts within the RTC, promoting its directional progression along the viral RNA. Similar to the conformational gating observed in the P-cluster of nitrogenase[72,73], redox-linked dynamics in RTC proteins may mediate crosstalk between enzymatic activities. Such redox switches could modulate RNA-binding affinities, trigger structural rearrangements, or both, providing a mechanism for temporal and spatial synchronization of the multiple functions embedded in the RTC, akin to redox-regulated control observed in DNA-processing enzymes[74].

An intriguing possibility is that SARS-CoV-2 may exploit Fe-S clusters to dynamically modulate its replication machinery in response to fluctuations in the redox landscape and metabolic state of the host cell[75]. Such a mechanism could provide a flexible means of tuning enzymatic activities to match intracellular conditions, thereby enhancing viral replication efficiency, adaptability, and pathogenic potential. If so, this redox-responsive regulation might represent an underappreciated aspect of the virus evolutionary success and a potential vulnerability that could be leveraged for therapeutic intervention. Accordingly, the Fe-S clusters represent a critical Achilles' heel of the virus, as they are highly sensitive to oxidative damage[38]. Targeting these cofactors offers a promising therapeutic approach to treat COVID-19. Small molecules with mild oxidative properties, such as TEMPOL, have shown significant potential in inhibiting viral replication in both in vitro and in vivo models[32]. These compounds likely act by inducing a shift from the stable $[Fe_4S_4]^{2+}$ to the often highly unstable $[Fe_4S_4]^{3+}$ state, which is prone to degradation[38], thereby inactivating the core component of the viral replication machinery.

Our studies identify multiple SARS-CoV-2 RTC components as Fe-S proteins, raising the possibility that redox-dependent conformational changes may regulate their activities and interactions within the complex and with host cellular components. The intrinsic sensitivity of Fe-S clusters to oxidative damage presents an opportunity to develop targeted therapies aimed at disrupting these cofactors required for viral replication.

## Methods

### Cell lines and cell culture conditions

HEK293 cells were purchased from ATCC (#CRL-1573™). Cells were propagated in Dulbecco's modified Eagle's medium (DMEM) with 4.5 g/L glucose, supplemented with 10% fetal bovine serum (FBS) and 2 mM glutamine at 37 °C and 5% $CO_2$ in a humidified incubator. Expi293F cells used for mammalian cell expression of the SARS-CoV-2 nsp14, nsp10 and nsp16 were purchased from ThermoFisher Scientific (#A14635). Cells were propagated in suspension in chemically defined, serum-free, protein-free, animal origin-free Expi293 Expression Medium at 37 °C and 8% $CO_2$ and subcultured according to the manufacturer's instructions. All cell lines were subjected to mycoplasma testing.

### Plasmids and transfection of mammalian cells

Recombinant expression of C-terminally Strep II-nsp14 in Expi293F and HEK293 cells was obtained from pLXV-EF1alpha-2xStrep-SARS-CoV-2-nsp14-IRES-Puro (Addgene Plasmid # 141380)[76]. Recombinant expression of C-terminally Strep II-nsp10 in Expi293F and HEK293 cells was obtained from pLXV-EF1alpha-2xStrep-SARS-CoV-2-nsp10-IRES-Puro (Addgene Plasmid # 141376)[76]. Recombinant expression of C-terminally His8-Flag-nsp10 in Expi293F and HEK293 cells was obtained from pcDNA5-FRT-TO-Nsp10-HF (Addgene Plasmid # 157708)[77]. Recombinant expression of C-terminally His8-Flag-nsp16 in Expi293F and HEK293 cells was obtained from pcDNA5-FRT-TO-Nsp16-HF (Addgene Plasmid # 157725)[77]. Point mutations into nsp14 and nsp10 were introduced using QuikChange II XL site-directed mutagenesis kit (Agilent Technologies #200522), following the manufacturer's instructions. All clones were verified for the insertion of the desired mutation(s) by Sanger sequencing at Eurofins USA. Plasmid

transfections into mammalian cells, except Expi293F cells (see below), were performed with Lipofectamine 2000 (ThermoFisher Scientific), according to standard procedures.

## List of mutagenesis primers

The list of mutagenesis primers used to generate the nsp14 and nsp10 variants reported in this study were purchased from Eurofins Genomics and can be found in the Supplementary Information.

## Protein production and purification

Expi293F cells at a final density of $3 \times 10^6$ viable cells/mL ($2.25 \times 10^9$ cells at the time of transfection) were inoculated in 800 ml of fresh Expi293$^{TM}$ Expression Medium supplemented with 300 µM L-cysteine and 40 µM FeCl$_3$ and transfected with a total of 840 µg of DNA plasmids encoding the proteins of interest (nsp14, nsp10, 1:1 nsp14/nsp10, or 1:1 nsp16/nsp10) which were combined with 48 ml of OptiMEM$^{TM}$ I Reduced Serum Medium (ThermoFisher Scientific) and incubated for 5 min at room temperature. The plasmid DNA mix was combined with a mix containing 2.6 ml Expifectamine$^{TM}$ 293 Reagent and 45 ml OptiMEM$^{TM}$ I Reduced Serum Medium and incubated at room temperature for additional 20 min prior to addition to the cell suspension. Forty-eight hours after transfection, cells were spun down and washed twice with cold PBS prior to being taken inside an argon recirculated glove box operated at <0.2 ppm $O_2$ for lysis. Cells were lysed in 100 ml of lysis buffer (150 mM NaCl, 50 mM Na-HEPES pH 7.4, 10% (v/v) glycerol, 3 mM MgCl$_2$, 2 mM TCEP, 0.5% NP-40, EDTA-free protease inhibitor cocktail) and homogenized by gently pipetting up and down until no cell clumps were visible. After clearing up the insoluble fractions by centrifuging the homogenate at 40,000 x g for 30 min, 4 ml of pre-equilibrated Strep II beads (IBA-Lifesciences) were added to the lysate. Recombinant protein complexes were immunoprecipitated for 4 h at 4 °C, before being packed into a column. The resin was washed with 10 column volumes of lysis buffer, followed by 20 column volumes of lysis buffer supplemented with 500 mM NaCl and 5 column volumes of wash buffer (100 mM Tris-Cl, 500 mM NaCl pH 8). Strep II tagged proteins (nsp14 and nsp10) were then eluted from the beads with elution buffer (100 mM Tris-Cl, 500 mM NaCl, 2.5 mM desthiobiotin, pH 8.0) for 2 h at room temperature. The eluates at 15 µM were concentrated to ~ 50 µM using either a 50 kDa or 10 KDa molecular weight cut-off filter. His8-Flag-nsp16 was co-expressed with nsp10, except when nsp10 was fully reconstituted with zinc. Flag-tagged nsp16 and nsp10 were competitively eluted using 3XFLAG peptide (Sigma #F4799) at a concentration of 100 µg/mL for 2 h at room temperature. The eluates, initially at 25 µM, were concentrated to ~100 µM using a 30 kDa molecular weight cut-off filter. To generate zinc-bound forms of nsp14/nsp10 and nsp10/nsp16 complexes, proteins were co-expressed in Expi293F mammalian cells and purified aerobically on the bench using the same protocol employed for anoxic purification. The lysis buffer (150 mM NaCl, 50 mM Na-HEPES pH 7.4, 10% (v/v) glycerol, 3 mM MgCl$_2$, 2 mM TCEP, 0.5% NP-40, and EDTA-free protease inhibitor cocktail) was supplemented with 100 µM ZnCl$_2$ to promote zinc incorporation. All steps were performed in the presence of atmospheric oxygen.

## Subcellular fractionation and immunoprecipitation (IP)

Subcellular fractionation into cytosol and organelles was done as previously described[48–50,78]. Briefly, cytosolic fractions from HEK293 or Expi293F cell pellets (~$10^9$ cells) were isolated from the organelles by permeabilizing cells with a buffer containing 0.1% digitonin in 210 mM mannitol, 20 mM sucrose, and 4 mM HEPES. The supernatant after centrifugation at 700 x g for 5 min, which contained cytosolic proteins, was spun down at 21,000 x g for 30 min. The supernatant after the centrifugation was supplemented with a 1:1 volume (v/v) of a buffer containing 25 mM Tris, 200 mM NaCl, 1 mM EDTA, and 1% NP-40 (pH 7.4) to obtain a protein concentration of ~1µg/µl. For the co-IP experiments in HEK293 cells, 500 µl of total cytosolic proteins were used for the immunoprecipitation (IP) of Strep II-tagged nsp14 and nsp10 using Strep II beads (IBA-Lifesciences). Equilibrated Strep II beads were added to the lysates and incubated overnight at 4 °C. Beads were recovered after 5 washes with lysis buffer and two washes with wash buffer (100 mM Tris-Cl, 500 mM NaCl pH 8). Proteins were eluted from the Strep II beads with Tris-Glycine pH 2.8 (150 µl/ IP sample) for 15 min at room temperature or for the native elution with elution buffer (100 mM Tris-Cl, 500 mM NaCl, 2.5 mM desthiobiotin, pH 8.0) for 2 h at room temperature. Flag-tagged proteins (nsp10 and nsp16) were incubated with anti-Flag M2 agarose beads (Sigma) overnight at 4 °C. Beads were recovered after 5 washes with lysis buffer and two washes with DPBS. Proteins were eluted from the beads with Tris-Glycine pH 2.8 (150 µl/ IP sample) for 15 min at room temperature or for the native elution with 3XFLAG peptide for 2 h at room temperature. The eluates were then analyzed by SDS PAGE and immunoblot.

## Iron incorporation assay

The $^{55}$Fe incorporation assays into the proteins of interest were performed as previously described[23], with some modifications. Expi293F cells were grown in expression medium in the presence of 1 µM $^{55}$Fe-Transferrin. Cytosolic extracts were subjected to immunoprecipitation to immunocapture the proteins of interest 48 h post-transfection. Samples collected after competitive elution (with 3XFLAG peptide at 100 µg/ml or with desthiobiotin) were analyzed by scintillation counting to assess $^{55}$Fe content. The background levels corresponding to $^{55}$Fe measurements on eluates after anti-FLAG immunoprecipitations on cytosolic extracts isolated from cells transfected with the empty vector, pcDNA5-FRT-TO-HF, or eluates after anti-Strep II IP on cytosolic extracts from cells transfected with the backbone vector, pLXV-EF1alpha-2 x Strep-IRES-Puro, were also included to account for nonspecific $^{55}$Fe amounts stochastically associated to the beads. When $^{55}$Fe levels incorporated into the proteins of interest were determined after knockdown of *NFS1*, cells were transfected twice with non-targeting or *NFS1*-directed siRNAs at a 48-h interval. At the time of the second transfection with si-RNAs, cells were co-transfected with the constructs encoding the proteins of interest. Cell lysates were analyzed 48 h after the second transfection.

## Si-RNA-mediated knockdown of *NFS1* in Expi293F cells

On-TARGET Plus siRNA pools against human *NFS1* (Cat. No. LQ-011564-01-0005) and the control nontargeting (NT si-RNAs) pool (Cat. No. D-001810-10-05) were purchased from Dharmacon. Knockdown of *NFS1* in Expi293F cells was achieved by transfecting cells twice with siRNAs at a 48 h interval using ExpiFectamine$^{TM}$ 293 Transfection Kit (ThermoFisher Scientific) according to manufacturer's instructions. At the time of the second transfection with si-RNAs, cells were co-transfected with the constructs encoding nsp14, nsp10 or nsp16. Cell lysates were analyzed 48 h after the second transfection by immunoblot to verify the efficiency of *NFS1* knockdown and by scintillation counting to measure $^{55}$Fe-incorporation into the proteins of interest in control (NT si-RNAs) or *NFS1* depleted (si-NFS1) cells.

## RNA substrate for the ExoN activity

To assay the ExoN activity of the nsp10/nsp14 complex, we used an RNA substrate consisting of a template strand with three initiating citidines followed by 31 nucleotides (nts). This strand also functioned as the nonscissile strand for ExoN. The complementary product strand, which served as the scissile strand for ExoN, was 29 nts in length and terminated with a cytidine-5'-monophosphate (CMP), creating a C-U mismatch at the 3'-end. To enable monitoring of exonuclease activity, the scissile strand was modified at the 5'-end to include a fluorophore, Alexa488N, a fluorescein-based dye derivative recognized for its photostability, strong fluorescence signal, and compatibility with RNA substrates.

3'-GCUUUCCCUUUUCCCCUUCCUCCUUCUUUCUUUCCC-5'
5'-CGAAAGGGAAAGGGGAAGGGGAAGAAAGAC-3'

The RNA substrate was purchased from Integrated DNA Technologies (IDT). It was annealed in 10 mM Tris-HCl (pH 7.5) and 100 mM KCl to make a partial duplex RNA, which was folded at 85 °C for 5 min, followed by snap-cooling on ice for 15 sec, finally allowing it to equilibrate at room temperature for 20 min.

### ExoN activity assays

The RNA substrate at a final concentration of 2 μM was incubated with the wild-type or variant SARS-CoV-2 nsp14/nsp10 complex ligating Fe-S or zinc cofactors, as indicated, at 50 nM in a reaction buffer containing 25 mM HEPES (pH 7.5), 50 mM NaCl, 1 mM TCEP, and 4 mM MgCl$_2$ at 37 °C. All reactions were carried out at 37 °C, and a time 0 sample was collected prior to incubation at 37 °C. The reactions were allowed to proceed for the indicated time points (0-30 min) and were stopped by adding an equal volume of 2 × TBE-Urea supplemented with 100 mM EDTA, followed by heating at 85 °C for 5 min. Reactions were resolved on 20% Urea gels in TBE buffer at 100 V for 2 h and imaged using a ChemiDoc MP imaging system (Bio-Rad). The nsp10 and nsp14 proteins used in the assay were co-expressed and purified under strictly anoxic conditions, except for the samples labeled "All Zn", which were co-expressed and purified under aerobic conditions and contained only zinc, as confirmed by ICP-MS analysis.

### Electrophoretic Mobility Shift Assays (EMSAs) to detect RNA binding by nsp14 and nsp10/nsp16

Electrophoretic mobility shift assays (EMSAs) were performed to assess RNA binding by nsp14 and the nsp10/nsp16 complex. The RNA template (sequence: [ppp] AUUAAAGGUU) was annealed in 10 mM Tris-HCl (pH 7.5) and 100 mM KCl. Annealing was carried out by heating the mixture to 85 °C for 3 min, snap-cooling on ice for 15 s, and then incubating at room temperature for 1 h. This substrate was used for EMSAs with nsp14.

For EMSAs with the nsp10/nsp16 complex, the same RNA substrate was enzymatically capped using the Vaccinia Capping System (New England Biolabs, Cat# M2080S) according to the manufacturer's protocol. The capped RNA (sequence: $^{7Me}$G[ppp] AUUAAAGGUU) was purified using the RNA Clean & Concentrator-5 kit (Zymo Research, Cat# R1014) and eluted in RNase-free water.

The assays were performed inside an argon recirculated glovebox maintained at <0.2 ppm O$_2$ to preserve the integrity of the Fe-S clusters in nsp14 and nsp10.

Binding reactions (20 μl) contained 25 mM HEPES (pH 7.4), 100 mM NaCl, 2 mM MgCl$_2$, 1 mM TCEP, 0.25 μg RNA, and increasing concentrations of purified proteins (0.25×, 0.5×, 1×, 1.5×, and 2× molar excess relative to RNA (at 0.5 nM)). Reactions were initiated by adding either nsp14 or the nsp10/nsp16 complex and incubated for 10 min at room temperature.

Reaction mixtures were quenched with the addition of 4 μl Stop Buffer (75% glycerol, 0.01% xylene cyanol, 0.01% bromophenol blue) and samples were resolved on 12% native acrylamide gels in 1 x TBE buffer at 200 V for 2.5 h. Gels were stained with toluidine blue to visualize RNA and imaged using a flatbed scanner.

To assess whether the Fe-S clusters in nsp14 and nsp10 are redox-active (i.e., capable of undergoing reduction or oxidation) nsp14 and nsp10/nsp16 complexes were treated with reducing agents (ascorbate or dithionite) or the oxidizing agent ferricyanide. Ascorbate was added at a 10-fold molar excess for nsp14 and up to 100-fold for nsp10/nsp16, dithionite at up to 10-fold excess, and ferricyanide at 1 molar equivalent.

### Methyltransferase (MTase) assays

All MTase reactions were performed using the MTase-Glo™ bioluminescence assay (Promega, V7601) in 96-well solid white assay plates.

RNA-1 (G(5')PPP(5')A) and Cap-0 (m$^7$G(5')ppp(5')) were used as substrates for the reactions catalyzed by nsp14 and the nsp16/nsp10 complex. The reactions were carried out at 37 °C for 30 min under the conditions specified in the main text and figure legends. Reactions were terminated by adding trifluoroacetic acid (TFA) to a final concentration of 1%. The MTase-Glo™ Detection Solution was then added, and the plate was mixed thoroughly before incubating for an additional 30 min. Luminescence was then recorded using a Tristar$^2$ LB942 Multimode Reader (Berthold Technologies).

### Modeling nsp16/nsp10 and nsp14

Coordinates for the nsp16/nsp10 complex bound to S-Adenosyl methionine and for nsp14 were taken from PDB IDs 6W4H[79] and 7QGI[80], respectively. Zn atoms at the metal-binding sites identified in the present study as Fe-S ligating centers were computationally replaced with Fe-S clusters. To allow for the insertion of the cluster into the Zn cavities, cofactor volumes were halved (this adjustment was transient, and the expected volume was restored during the subsequent optimization). All missing hydrogen atoms were added with VMD[81]. For each complex, a total of three simulation cells were prepared - one with both cofactors oxidized, and two with only one of the two cofactors reduced. All six systems were solvated in a box with a padding of 15 Å around the complex, and sodium ions were added to neutralize the overall charge. After 50,000 minimization steps, during which all force constants for the Fe-S cluster angles were increased tenfold to enforce the desired geometry, the systems were equilibrated in NAMD[82] through 6 ns of molecular dynamics (MD) in the NPT ensemble ($T = 310$ K and $p = 1$ atm), followed by 24 ns in the NVT ensemble.

### Calculating the reduction potentials

Reduction potentials of the Fe-S clusters were calculated by the linear response approximation within the thermodynamic integration method[83,84]. Vertical energy gaps at the geometries of the reduced and oxidized potential energy surfaces were obtained by single point quantum mechanics/molecular mechanics calculations (QM/MM) in CP2K[85]. For each system, we used 100 configurations saved from a 50 ns production run simulated by NAMD. Reduction potentials of transition-metal complexes obtained via QM/MM can systematically deviate from the experimental values by several hundred mVs[86,87]. To account for the offset, we adopted the ferredoxin from *Thermotoga maritima* (PDB ID: 1VJW[88]) as a reference system. Our calculated reduction potential for its [Fe$_4$S$_4$] cluster was -1.08 V, against the experimental value of -0.42 V[89], revealing a systematic offset of +0.66 V, which we applied as a correction to all our results to mitigate methodological biases.

### Simulation details

The MM interactions were modeled by the CHARMM36 force field[90], with the parameters for the Fe-S clusters derived by Chang et al. and refined by McCullagh et al.[91,92]. A cutoff radius of 12 Å was used for all nonbonded interactions. The MD time step was 2 fs. The QM region, described by density functional theory (DFT), included the Fe-S cluster and the side chains of its coordinating ligands up to the beta carbons, capped with link hydrogen atoms. We employed the PBE functional[93], the TZVP-MOLOPT basis set[94], and the GTH pseudopotentials proposed by Goedecker, Teter and Hutter[95]. DFT calculations were performed with a plane-wave cutoff of 400 Ry and a target accuracy for the self-consistent field convergence of $1.0 \times 10^{-5}$. The antiferromagnetic (AF) coupling of the Fe atoms was treated by the broken symmetry approach implemented in CP2K. To balance accuracy and computational resources, we calculated energy gaps only for one of the six AF configurations of the oxidized state, selecting one of the two Fe$^{3+}$ ions as the electron acceptor. Preliminary tests estimated that the variability in the reduction potentials, arising both from different AF

couplings and the choice of $Fe^{3+}$ ion as the electron acceptor, corresponds to -0.1 V.

## Mass spectrometry analysis

Proteins (~10 μg per sample) were run on SDS-PAGE and stained with Coomassie Blue G-250. The gel bands were excised and washed overnight in 50% methanol with 10% acetic acid. Proteins were reduced using 5 mM Tris(2-carboxyethyl) phosphine hydrochloride at room temperature for 1 h, then alkylated with 5 mM N-ethylmaleimide (NEM) for 10 min and digested with trypsin (Promega) 1:20 (w/w) at 37 °C for 18 h. Tryptic digests were extracted from the gel and cleaned with an Oasis HLB microplate (Waters). The desalted peptides were injected into a Dionex UltiMate 3000 RSLCnano HPLC instrument (Thermo-Fisher Scientific) with an ES802 nanocolumn (ThermoFisher Scientific). The column temperature was set at 45 °C. Mobile phase A and B (MPA, MPB) contained 0.1% formic acid in water and 0.1% formic acid in acetonitrile, respectively. The peptides were eluted at a flow rate of 300 nL/min using the following gradients: 3% to 22% MPB for 88 min, 22% to 33% MPB for 10 min, 33% to 80% MPB for 6 min. Thermo Orbitrap Fusion mass spectrometer (ThermoFisher Scientific) was used for data acquisition. The LC-MS/MS data were acquired in data-dependent mode. The MS1 scans were performed in orbitrap with a resolution of 120 K at 200 m/z with a mass range of 400−1500 m/z and an automatic gain control (AGC) value of $2 \times 1e5$. The quadrupole isolation window was 1.6 m/z. Ions with an intensity $>1 \times 1e4$ were fragmented by HCD method with collision energy fixed at 35%. The MS2 scans were conducted in ion trap with and AGC target of $3 \times 1e4$.

The Proteome Discoverer software version 2.4 was used for protein identification and quantitation. Raw data were searched against Uniprot Human Database. The mass tolerances for precursor and fragment were set to 5 ppm and 0.6 Da, respectively. Up to 2 missed cleavages were allowed for trypsin digestion. NEM on cysteines was set as fixed modification. Variable modifications include Oxidation (M), Met-loss (Protein-N-term) and Acetyl (Protein N-term). Peptides were validated based on $q$-values using percolator algorithm. The search results were filtered by a false discovery rate (FDR) of 1% at the protein level. The protein abundances were calculated by summing the abundance of the connected peptides. The protein ratios were calculated by dividing the protein abundance values between the eluates after immunoprecipitation of Strep II- nsp14 (nsp14, samples; $n = 4$) and those of the IgG immunoprecipitations (control, samples; $n = 4$). Proteins were considered detected only when they were found in at least 3 out of 4 replicates. ANOVA (Individual Proteins) method was used for statistical analyses. Because the ratios were calculated without normalization and imputation, if a target protein was not detected in either the 'nsp14' or 'control' group samples, a fold change value was calculated and reported without $p$-value.

## Ferrozine based colorimetric assay to determine iron stoichiometries of nsp14 and nsp10 WT and variants

The ferrozine based colorimetric assay (Sigma-Aldrich, Cat. No.: MAK025-1KT) was used to determine the concentration of iron ($Fe^{2+}$) in preparations of purified nsp14 and nsp10 WT and variants each at 50 μM protein concentration, according to the manufacturer's instructions. In this assay, the iron released by the addition of an acidic buffer was reduced to measure both $Fe^{2+}$ and $Fe^{3+}$ and colorimetrically determined at 593 nm after reaction with a chromogen proportional to the iron present in the samples. To accurately determine protein concentrations for nsp14 and nsp10 WT and variants avoiding systematic over or under-estimations inherent to the routinely used colorimetric methods that are based on the absorbance of a standard protein (usually bovine serum albumin), protein concentrations determined with the method of Bradford were corrected with respect to the BSA standard by performing amino acid analysis (Alphalyse

Inc.)[96]. Iron concentrations in nsp14 WT, nsp14$^{LFK-AAA}$, nsp14$^{E191A}$, nsp14$^{C210S-C226S-H229S}$, nsp14$^{C261S-H264S-C279S}$, nsp14$^{C477S-C484S-H487S}$ each at 50 μM were $375.2 \pm 54.5$ μM, $15 \pm 2.5$ μM, $360.5 \pm 51.9$ μM, $51.3 \pm 12.5$ μM, $25.2 \pm 11.3$ μM, and $65.1 \pm 14.3$ μM respectively, corresponding to $7.5 \pm 1.9$ iron/protomer for nsp14 WT, $0.3 \pm 0.1$ for nsp14$^{LFK-AAA}$, $7.2 \pm 1.0$ for nsp14$^{E191A}$, $1.0 \pm 0.2$ for nsp14$^{C210S-C226S-H229S}$, $0.2 \pm 0.2$ for nsp14$^{C261S-H264S-C279S}$, and $1.0 \pm 0.3$ for nsp14$^{C477S-C484S-H487S}$. Iron concentrations for nsp10 WT, nsp10$^{C74S-C77S-H83S}$ and nsp10$^{C120S-C128S-C130S}$ each at 50 μM were $340 \pm 27$ μM, $23 \pm 21$ μM and $35 \pm 18$ μM, respectively, corresponding to $6.8 \pm 0.5$ iron/protomer for nsp10 WT, $0.5 \pm 0.4$ for nsp10$^{C74S-C77S-H83S}$ and $0.7 \pm 0.4$ for nsp10$^{C120S-C128S-C130S}$.

## Inductively coupled plasma mass spectrometry (ICP-MS)

Total iron and zinc content in the samples were measured by ICP-MS (Agilent model 7900). For each sample, 200 μL of concentrated trace-metal-grade nitric acid (Fisher) was added to 200 μL of sample taken in a 15 mL Falcon tube. The tubes were then incubated at 85 °C in an oven for overnight digestion. Each sample was diluted to a total volume of 4 mL with deionized water, and analyzed by ICP-MS.

## Ultraviolet-visible (UV-vis) absorption spectroscopy and amino acid analysis (AAA)

UV-vis spectra were acquired for anoxically purified proteins in sealed, air-tight cuvettes using a NanoDrop spectrophotometer (Thermo-Fisher Scientific) with the elution buffer containing either 100 mM Tris-Cl, 500 mM NaCl, 2.5 mM desthiobiotin, pH 8.0 or 100 mM Tris-Cl, 500 mM NaCl, 100 μg/ml 3XFLAG peptide as blank. Amino acid analysis was performed by Alphalyse Inc. to precisely quantify the purified proteins.

## EPR spectroscopy

For EPR analyses, 200 μL aliquots of anoxically purified nsp14 or nsp10, each at a final concentration of 50 μM, were transferred to EPR tubes. To test whether the Fe-S cluster(s) could be reduced and to characterize the reduced cluster(s), the proteins were incubated with 0.5 mM sodium dithionite for 15 min prior to freezing. Continuous-wave EPR (CW-EPR) spectra were acquired on a Bruker ELEXSYS-II E580 X-band spectrometer. The temperature was maintained at 10 K using a CF935O cryostat (Oxford Instruments) and an ITC503 temperature controller (Oxford Instruments). The spectrometer was operated through Bruker's Xepr software on a dedicated workstation. The modulation amplitude was set to 5 G, the microwave frequency was 9.439 GHz, and the microwave power was 20 mW. Spectral simulations were performed using the EasySpin package 6.0.6[97].

## EDTA treatment and reconstitution with zinc of wild type nsp14 and nsp10 for ICP-MS analyses

To obtain metal-free (apo-) nsp14 and nsp10, the proteins purified aerobically were treated with 10 mM EDTA for 1 h and passed through a PD G-25 column (GE Healthcare). For full reconstitution with zinc, which yielded a protein containing 3 zinc ions per protomer for nsp14 and 2 zinc ions per protomer for nsp10, aerobically purified proteins were incubated with 5 mM dithiothreitol and 10 equivalents of $ZnCl_2$ at room temperature for 2 h and subsequently passed through a PD G-25 column (GE Healthcare). Metal composition of the aerobically purified proteins fully reconstituted with zinc was confirmed by ICP-MS.

## Antibodies

Antibodies in this study were as follows: anti-HSC20 westerns were performed either with a custom-made antibody raised against the whole protein (Genscript) or with a commercial antibody (Sigma #HPA018447). Anti-CIAO1 (sc-374498), PCBP1 (sc-393075), and NFS1 (sc-81107) were from Santa Cruz Biotechnology. Anti-HSPA9 (HPA000898) was from Sigma. Anti-FAM96B (20108-1-AP) and

MMS19 (16015-1-AP) were from Proteintech. Anti-FLAG antibody was from Origene (TA50011). Anti-Strep II was from Qiagen (1023944). Anti-BOLA2 was from Bethyl Laboratories (A305-890A-M). Anti α-Tubulin was obtained from Sigma (T9026). Primary antibodies were used at a 1:1,000 dilution and incubated overnight at 4 °C

Uncropped images of all gels and blots for the main and Supplementary Figures are provided at the end of the Supplementary Information.

Molecular weights are indicated in kilodaltons (kDa) for all gels and western blot panels (Figs. 1a, c–e, g, 2b, c, 4b, 6c, f).

### Graphical illustration
Figure 3b was created with BioRender.com under an NIH institutional license.

### Statistical analyses
Where applicable, pairwise comparisons between two groups were analyzed using the two-way ANOVA and Sidak's multiple comparisons test. All tests were performed with GraphPad Prism 10.2.3, and data were expressed as mean ± standard deviation (SD).

### Reporting summary
Further information on research design is available in the Nature Portfolio Reporting Summary linked to this article.

## Data availability
All data needed to evaluate the conclusions of the paper are present in the main text and supplementary information. Source data are provided with this paper. The mass spectrometry data have been deposited to ProteomeXchange PXD060605. Source data are provided with this paper.

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

## Acknowledgements

The authors thank the *Eunice Kennedy Shriver* National Institute of Child Health and Human Development for support. U.T. and N.M. acknowledge funding from the Royal Society International Exchange Scheme (IES\R3\243122), which facilitated collaborative research contributing to this study. This work was funded by the Intramural Research Program of the National Institutes of Health (1ZIA HD008814- Mammalian iron-sulfur cluster biogenesis) and by the National Institutes of Health extramural grant R35 GM127079 awarded to C.K. Via U.T.'s membership of the UK HEC Materials Chemistry Consortium, which is funded by EPSRC (EP/X035859), this work used the ARCHER2 UK National Supercomputing Service (http://www.archer2.ac.uk).

## Author contributions

N.M. designed the project, wrote the manuscript and designed and performed most of the experiments, except as follows: U.T. performed the QM/MM simulations, Y.L. performed mass spectrometry sample preparation and analysis. N.M., T.A.R., U.T., Y.L., J.M.B. Jr., and C.K. analyzed the data. All authors revised the manuscript.

## Funding

## Competing interests

The authors declare no competing interests.
