## [Transparent Peer Review file · Nature Communications]

Iron-sulfur clusters in SARS-CoV-2 exoribonuclease and methyltransferase complexes: relevance for viral genome proofreading and capping

Corresponding Author: Dr Nunziata Maio

Version 0:

Reviewer comments:

Reviewer #1

(Remarks to the Author)

This work builds on prior studies by this research team that involves the binding of Fe-S clusters in non-structural proteins (nsp's) involved in the replication-transcription of activities necessary to achieve propagation of SARS-Cov-2 published in Science and PNAS. The current work specifically involves analysis of the nsp14/nsp10 complex having 3'-5' exoribonuclease activity as well as nsp14 (alone) having N7-methyltransferase SARS-CoV-2 genomic capping activity. In aggregate the authors establish that nsp14 and nsp10 individually contain [4Fe-4S] clusters which are required to sustain a high level of their corresponding activities. Evidence for the binding of [4Fe-4S] clusters is provided by biophysical characterization of the cognate clusters, the identification of a cluster targeting signature, and the co-immunoprecipitation of factors known to be involved in Fe-S cluster formation and delivery with nsp10/nsp14. All of the work that was used to establish these basic features is experimentally sound and convincing.

The authors speculate at length on the possible involvement of the associated [Fe-S] clusters in redox activities that could modulate SAR-CoV-2 propagation but there is no experimental basis to warrant that suggestion. The authors show that the as-isolated proteins replete with Fe-S clusters show no EPR signature, but treatment with the powerful reducing agent, dithionite, yields an EPR signature characteristic of [4Fe-4S] clusters. What is not shown is whether or not the species can be re-oxidized to establish a redox cycling capacity. It is also unexpected that the EPR signatures of the reduced species of nsp14 and nsp10 are almost identical. All of the careful quantitative metal analyses provide good evidence that 2 [4Fe-4S] clusters are present in nsp10 and nsp14 but this possibility has not been explored by EPR. If there are indeed two clusters present in nsp10 and nsp14 it might be expected that this could be conclusively established by power and temperature-dependence measurement of the spectra shown in Figure 2e. Finally, because EPR signatures can be elucidated by the addition of reducing equivalents it would seem reasonable that the midpoint potentials could be determined experimentally rather than computationally. Another concern about the suggestion that the [4Fe-4S] clusters have a redox, rather than only a structural, role is that the relevant activities (although low) persist in the absence of the cognate clusters. The objection here is not whether or not it is reasonable to suggest that there could be a redox function for the associated [4Fe-4S] clusters, but rather the authors proceed to discuss that possibility as an established fact.

The last paragraph in the introduction describes the results and conclusions and, therefore, not necessary.

Reviewer #2

(Remarks to the Author)

Maio et al. show that SARS-CoV-2 nsp14 and nsp10 expressed in mammalian cells both contain Fe-S clusters occupying the metal binding sites previously observed to coordinate Zn, similarly to nsp12 and nsp13 as reported earlier by the same authors. They found that Fe-S-bound nsp14 has better N7 methyl transferase activity compared to Zn-bound nsp14 whereas it has comparable ExoN activity to Zn-bound nsp14, and Fe-S-bound nsp10 activates the nsp16 2'-O methyl transferase activity better than the Zn-bound nsp10. The authors propose that the Fe-S clusters in SARS-CoV-2 RTC may mediate conformational changes to help coordinate the activities of these proteins to facilitate virus replication. Overall, the biochemical data showing the presence of Fe-S clusters in nsp14 and nsp10 are convincing and this work provides new

insights into the structure and function of these important viral proteins. However, the authors should address the following points.

1. Nsp16 is used as a negative control in demonstrating the selective interaction of nsp14 or nsp10 with Fe-S biogenesis factors. But nsp16 does not have a Zn site. It would be more meaningful to use proteins with known Zn (and not Fe-S) sites for this purpose.
2. It is stated that “The nsp14/nsp10 complex was obtained by co-expressing the two proteins (wild-type or variants, as indicated) in Expi293F mammalian cells and copurifying them under anoxic conditions. For fully zinc-reconstituted nsp14 and nsp10, the proteins were individually expressed, purified under ambient oxygen conditions, and reconstituted with zinc, as outlined in the methods section. The metal composition of the proteins was confirmed by ICP-MS. “
Co-expression with nsp10 is likely to affect the folding of nsp14 and its activity. For activity comparisons, the authors should make a head-to-head comparison between proteins from the same source, purified either anoxically or aerobically.
3. The quality of the ExoN activity data is poor, and the description of the results is not quantitative. The authors state that the internal FAM label was used to enable real-time monitoring of exonuclease activity, but no real-time data are provided. It is stated that Fe-S-nsp14 and zinc-reconstituted nsp14 were equally efficient in cleaving the RNA substrate, but they look somewhat different on the gels shown. It's also unclear why the products look so different between the two gels in Fig. 3a.
4. The way the LFK variant data are currently presented is confusing. It appears that the LFK variant is inactive in either the exonuclease or methyltransferase assay for reasons unrelated to Fe-S loading. The loss of the exonuclease activity could be due to the mutations near the N-terminus of nsp14, which is close to the ExoN active site, but it is puzzling that the 3xAla mutations kill the methyltransferase activity of the C-terminal domain. It should be at least as active as Zn-bound nsp14.
5. The discussion about the role of the Fe-S clusters is vague. The authors state that multiple Fe-S clusters in the RTC may enable conformational changes that facilitate the coordination of the activities of the proteins that ligate them, enhancing RNA synthesis, and ensuring the proper progression of the replication cycle. However, it is unclear what the nature of the conformational changes is and how Fe-S clusters are involved in the process.

Version 1:

Reviewer comments:

Reviewer #1

(Remarks to the Author)

The manuscript has been substantially revised in accordance with the comments provided by both reviewers. The clear explanations provided by the authors to justify/clarify their conclusions is appreciated.

Reviewer #2

(Remarks to the Author)

My comments have been addressed.

Eunice Kennedy Shriver
National Institute of Child Health
And Human Development (NICHD)
Metals Biology and Molecular Medicine Group
Section on Human Iron Metabolism
Building 35A, Room 2D-824
35A Convent Drive, MSC 3758
Bethesda, MD 20892-3758, USA
T: 301-496-7060; F: 301-402-0078
E-mails: maion@mail.nih.gov; rouault@mail.nih.gov

June 20th, 2025

To the Reviewers,

We are grateful for Their thoughtful and constructive feedback.

Below we provide a detailed point-by-point response to Their comments, with all relevant revisions and additions now incorporated into the manuscript.

Reviewer #1 (Remarks to the Author):

This work builds on prior studies by this research team that involves the binding of Fe-S clusters in non-structural proteins (nsp's) involved in the replication-transcription of activities necessary to achieve propagation of SARS-Cov-2 published in Science and PNAS. The current work specifically involves analysis of the nsp14/nsp10 complex having 3'-5' exoribonuclease activity as well as nsp14 (alone) having N7-methyltransferase SARS-CoV-2 genomic capping activity. In aggregate the authors establish that nsp14 and nsp10 individually contain [4Fe-4S] clusters which are required to sustain a high level of their corresponding activities. Evidence for the binding of [4Fe-4S] clusters is provided by biophysical characterization of the cognate clusters, the identification of a cluster targeting signature, and the co-immunoprecipitation of factors known to be involved in Fe-S cluster formation and delivery with nsp10/nsp14. All of the work that was used to establish these basic features is experimentally sound and convincing.

The authors speculate at length on the possible involvement of the associated [Fe-S] clusters in redox activities that could modulate SAR-CoV-2 propagation but there is no experimental basis to warrant that suggestion. The authors show that the as-isolated proteins replete with Fe-S clusters show no EPR signature, but treatment with the powerful reducing agent, dithionite, yields an EPR signature characteristic of [4Fe-4S] clusters. What is not shown is whether or not the species can be re-oxidized to establish a redox cycling capacity. It is also unexpected that the EPR signatures of the reduced species of nsp14 and nsp10 are almost identical. All of the careful quantitative metal analyses provide good evidence that 2 [4Fe-4S] clusters are present in nsp10 and nsp14 but this possibility has not

been explored by EPR. If there are indeed two clusters present in nsp10 and nsp14 it might be expected that this could be conclusively established by power and temperature-dependence measurement of the spectra shown in Figure 2e. Finally, because EPR signatures can be elucidated by the addition of reducing equivalents it would seem reasonable that the midpoint potentials could be determined experimentally rather than computationally. Another concern about the suggestion that the [4Fe-4S] clusters have a redox, rather than only a structural, role is that the relevant activities (although low) persist in the absence of the cognate clusters. The objection here is not whether or not it is reasonable to suggest that there could be a redox function for the associated [4Fe-4S] clusters, but rather the authors proceed to discuss that possibility as an established fact.

We thank the Reviewer for these insightful comments. In response, we have conducted additional experiments to directly test whether the Fe-S clusters in nsp14 and nsp10 are redox-active and whether their redox state influences RNA binding. These new data are now described in the final paragraph of the Results section. To investigate the redox responsiveness of the clusters, we treated nsp14 and nsp10/nsp16 with chemical reductants (ascorbate, dithionite) and an oxidant (ferricyanide) to shift the cluster valence from the resting [2+] state toward reduced [1+] or oxidized [3+] states, and assessed their binding to RNA-1 (nsp14) or Cap-0 structures (nsp10/nsp16). Ascorbate acts as a mild reductant (reduction potential -0.081 V vs SHE at pH 7(1)), dithionite as a strong reductant (reduction potential -0.66 V(2)), and ferricyanide as a potent oxidant (reduction potential +0.36 V vs SHE at pH 7(3, 4)); all three are widely used to modulate Fe-S cluster redox states in vitro. Attempts to oxidize the Fe-S clusters in nsp14 or nsp10 using ferricyanide resulted in cluster degradation, as reflected by the disappearance of the characteristic absorbance peak in UV-visible spectra, indicating that the clusters cannot undergo a stable [2+/3+] redox transition (Supplementary Fig. 5). While ferricyanide treatment led to loss of the characteristic Fe-S UV-visible absorbance, suggesting cluster degradation rather than stable oxidation to the [3+] state (Supplementary Fig. 5), the clusters could be reduced. Specifically, nsp14 clusters were readily reduced by ascorbate in a concentration-dependent manner, whereas those in nsp10 required stronger reductants, consistent with their lower calculated reduction potentials (Fig. 5a, 5b).

We probed experimentally the midpoint potentials of the clusters in nsp14 and nsp10. The Fe-S clusters in nsp14 were susceptible to reduction by ascorbate, with a two-fold molar excess resulting in a 40% reduction and a ten-fold excess achieving 77% reduction (Fig. 6a). In contrast, the clusters in nsp10 were much less efficiently reduced by ascorbate: only 16% reduction was observed with a ten-fold excess, and 32% with a hundred-fold excess (Fig. 6d). This differential reactivity suggests that the clusters in nsp10 are more electronegative, displaying a lower affinity for electrons and thus a reduced propensity to accept electrons from ascorbate. These findings align with the calculated redox potentials, which indicate lower reduction potentials (i.e., less favorable reduction) for nsp10 clusters compared to those in nsp14 (Fig. 5a, 5b). Dithionite, a much stronger reductant, was markedly more effective. A two-fold excess of dithionite reduced the clusters in nsp14 by 90%, while a ten-fold excess reduced the clusters in nsp10 by 80%. This demonstrates that, despite their lower reduction potential, the clusters in nsp10 can be reduced efficiently with a sufficiently strong reductant, consistent with their calculated redox properties and the EPR spectra (Fig. 2e), showing a more intense signal for nsp14 upon reduction, further corroborating the greater reducibility of its clusters compared to nsp10. We next examined whether redox-induced changes in Fe-S cluster oxidation states affected RNA binding by nsp14 and the nsp10/nsp16 complex. A ten-fold excess of ascorbate, which effectively reduced the clusters in nsp14, markedly impaired its binding to RNA-1

(Fig. 6b, 6c). In contrast, the same treatment had no effect on the RNA binding activity of nsp10/nsp16 (Fig. 6e, 6f), consistent with the limited reduction of nsp10 clusters under these conditions (Fig. 6d). However, treatment with a ten-fold excess of dithionite, which robustly reduced clusters in both proteins, led to a profound loss of RNA-binding activity by both nsp14 and nsp10/nsp16 (Fig. 6b, 6e). Notably, zinc-substituted forms of nsp14 and nsp10/nsp16 exhibited similarly impaired RNA binding (Fig. 6b, 6e). Together, these findings suggest that the redox state of the Fe-S clusters affects the RNA-binding function, and that reduction from the resting [2+] to the [1+] state impairs this activity.

Regarding the similarity of the EPR signatures of reduced nsp14 and nsp10: while unexpected, this outcome likely reflects the dominant electronic characteristics of the $[\text{Fe}_4\text{S}_4]^{1+}$ state, which tend to overshadow minor variations in ligation geometry (e.g., Cys₄ vs. Cys₃His coordination), particularly under low-temperature and high-dithionite conditions. As such, EPR g-values and line shapes are not always sensitive enough to resolve individual cluster environments unless large structural perturbations or advanced labeling strategies are employed. EPR alone cannot assign signals to specific clusters within proteins harboring multiple Fe-S centers. Attempts to generate single-cluster variants via mutagenesis led to complete loss of Fe-S incorporation (Fig. 2f, 2g), precluding site-specific spectroscopic interrogation. This limitation, combined with the lack of spatial resolution in EPR, hinders cluster-specific signal assignment. Nevertheless, our combined metal quantification and mutational data support the presence of two clusters in each protein, and the EPR spectra likely represent a convolution of signals from both.

To address the Reviewer's final point, we experimentally validated differences in redox behavior using UV-vis and RNA-binding assays under defined conditions, in agreement with our QM/MM-derived redox potential estimates. Our findings show that redox changes in the Fe-S clusters of nsp14 and nsp10 markedly influence RNA binding: reduction of the clusters diminishes nsp14 binding to RNA-1 and nsp10/nsp16 binding to Cap-0 structures. These effects suggest a regulatory role for the clusters in RNA engagement rather than direct participation in catalysis, which may explain the persistence of low-level activity in the absence of the cognate clusters. We have revised the text to clearly distinguish speculation from established findings and have substantially edited the Discussion to avoid overstating the redox-related role of the clusters.

The last paragraph in the introduction describes the results and conclusions and, therefore, not necessary.

We agree and have deleted the final paragraph of the Introduction.

Reviewer #2 (Remarks to the Author):

Maio et al. show that SARS-CoV-2 nsp14 and nsp10 expressed in mammalian cells both contain Fe-S clusters occupying the metal binding sites previously observed to coordinate Zn, similarly to nsp12 and nsp13 as reported earlier by the same authors. They found that Fe-S-bound nsp14 has better N7 methyl transferase activity compared to Zn-bound nsp14 whereas it has comparable ExoN activity to Zn-bound nsp14, and Fe-S-bound nsp10 activates the nsp16 2'-O methyl transferase activity better than the Zn-bound nsp10. The authors propose that the Fe-S clusters in SARS-CoV-2 RTC may

mediate conformational changes to help coordinate the activities of these proteins to facilitate virus replication. Overall, the biochemical data showing the presence of Fe-S clusters in nsp14 and nsp10 are convincing and this work provides new insights into the structure and function of these important viral proteins. However, the authors should address the following points.

1. Nsp16 is used as a negative control in demonstrating the selective interaction of nsp14 or nsp10 with Fe-S biogenesis factors. But nsp16 does not have a Zn site. It would be more meaningful to use proteins with known Zn (and not Fe-S) sites for this purpose.

We appreciate the Reviewer's comment and the opportunity to clarify this point. We did not use nsp16 as a negative control in our experiments, nor did we describe it as such in the manuscript. Rather, nsp16 was included to probe its potential interaction with the Fe-S biogenesis machinery, given its functional interplay with nsp10 and its role in the viral replication complex. This allowed us to assess whether components that interact with nsp10 might also associate with its binding partner in a complex.

To ensure specificity, all co-immunoprecipitation experiments included an IgG IP as a negative control. We chose to focus on non-structural proteins from SARS-CoV-2 for these interaction studies to preserve relevance to the viral replication-transcription machinery and avoid confounding variables associated with heterologous host protein expression. We agree that including a structurally comparable viral protein with a known zinc-binding site and no Fe-S coordination could, in principle, further support the selectivity of the interactions observed. However, selecting such a control is not straightforward; several viral proteins previously presumed to bind only zinc have, upon more detailed investigation by our group and others, been shown to ligate Fe-S clusters. Given this evolving landscape and within the scope of this study, we prioritized comparisons among replication-associated SARS-CoV-2 nonstructural proteins. This approach allowed us to assess specificity within the context of the viral replication machinery itself.

2. It is stated that “The nsp14/nsp10 complex was obtained by co-expressing the two proteins (wild-type or variants, as indicated) in Expi293F mammalian cells and copurifying them under anoxic conditions. For fully zinc-reconstituted nsp14 and nsp10, the proteins were individually expressed, purified under ambient oxygen conditions, and reconstituted with zinc, as outlined in the methods section. The metal composition of the proteins was confirmed by ICP-MS. “ Co-expression with nsp10 is likely to affect the folding of nsp14 and its activity. For activity comparisons, the authors should make a head-to-head comparison between proteins from the same source, purified either anoxically or aerobically.

We thank the Reviewer for this very valid point. We have now purified the nsp14/nsp10 complex following co-expression in Expi293F mammalian cells both anoxically (as done previously) and aerobically following the same protocol. The metal content of the aerobically purified complex (without additional zinc reconstitution) is included in Fig. 2f and 2g, and these preparations were used in newly performed ExoN activity assays shown in Fig. 3 and Supplementary Fig. 3. To generate the zinc-bound forms of the nsp14/nsp10 and nsp10/nsp16 complexes, proteins were co-expressed in Expi293F mammalian cells and purified aerobically on the bench following the same workflow used under anoxic conditions. The lysis buffer (150 mM NaCl, 50 mM Na-HEPES pH 7.4, 10% (v/v)

glycerol, 3 mM MgCl₂, 2 mM TCEP, 0.5% NP-40, and EDTA-free protease inhibitor cocktail) was supplemented with 100 μM ZnCl₂ to promote zinc incorporation. All steps were performed under ambient oxygen. While we agree that co-expression can influence folding and complex formation, and therefore appreciate the opportunity to strengthen this aspect of the study, we also note that several prior studies have performed biochemical and structural analyses using individually expressed and purified nsp14 and nsp10, which were then allowed to form a complex in vitro (e.g., PMID: 34942146; PMID: 34198326; PMID: 33705595).

3. The quality of the ExoN activity data is poor, and the description of the results is not quantitative. The authors state that the internal FAM label was used to enable real-time monitoring of exonuclease activity, but no real-time data are provided. It is stated that Fe-S- nsp14 and zinc-reconstituted nsp14 were equally efficient in cleaving the RNA substrate, but they look somewhat different on the gels shown. It's also unclear why the products look so different between the two gels in Fig. 3a.

We thank the Reviewer for this feedback and acknowledge the suboptimal quality of the initial ExoN activity data. The RNA oligonucleotides used in the original assays were ordered from IDT, but after multiple delays, the company was unable to deliver long oligos that met their quality control standards. Despite this, they provided the oligos we used in the assay, which resulted in some smearing visible at time zero.

To address this issue, we redesigned the RNA substrate to improve assay quality and reliability. The new substrate consists of a template strand that begins with three cytidines followed by 31 nucleotides, serving as the non-scissile strand (Fig. 3a). The complementary scissile strand is 29 nucleotides long, terminating in a cytidine-5'-monophosphate (CMP), which creates a C-U mismatch at the 3' end. The 5' end of the scissile strand is labeled with Alexa488, a fluorescein-based dye chosen for its strong fluorescence, photostability, and compatibility with RNA substrates. This labeling enables fluorescence-based monitoring directly tied to exonuclease progression.

We believe that these modifications have improved the quality of the data, as shown in the revised Fig. 3a and Supplementary Fig. 3. Both Fe-S-bound and zinc-bound nsp14/nsp10 complexes exhibit comparable activity on the mismatched RNA substrate.

Regarding the differences in product patterns between the two gels in the original Fig. 3a, these arose from the bottom gel being run for approximately one hour longer than the top gel, which affected band resolution. This discrepancy has been eliminated in the updated figures.

We hope these improvements and clarifications fully address the Reviewer's concerns.

4. The way the LFK variant data are currently presented is confusing. It appears that the LFK variant is inactive in either the exonuclease or methyltransferase assay for reasons unrelated to Fe-S loading. The loss of the exonuclease activity could be due to the mutations near the N- terminus of nsp14, which is close to the ExoN active site, but it is puzzling that the 3xAla mutations kill the methyltransferase activity of the C-terminal domain. It should be at least as active as Zn-bound nsp14.

We thank the Reviewer for this insightful comment. We have clarified this point in the revised main text, noting that although the LFK-to-AAA substitution is located in the N-terminal domain of nsp14 (Supplementary Fig. 1a), it disrupts the incorporation of both Fe-S clusters, including the one in the C-terminal domain required for N7-methyltransferase activity, as assessed by ICP-MS analysis (Fig. 2f-h). The lack of Fe-S coordination likely causes broader structural destabilization and misfolding, resulting in loss of function in both enzymatic domains. Thus, the observed enzymatic inactivity of the LFK mutant is consistent with loss of Fe-S incorporation and impaired folding rather than direct effects of the amino acid substitutions on either catalytic site.

5. The discussion about the role of the Fe-S clusters is vague. The authors state that multiple Fe-S clusters in the RTC may enable conformational changes that facilitate the coordination of the activities of the proteins that ligate them, enhancing RNA synthesis, and ensuring the proper progression of the replication cycle. However, it is unclear what the nature of the conformational changes is and how Fe-S clusters are involved in the process.

We appreciate the Reviewer's comment regarding the role of Fe-S clusters in facilitating conformational changes within the replication-transcription complex (RTC). While our data demonstrate that Fe-S clusters are present in nsp14 and nsp10, and that their redox state modulates RNA binding and methyltransferase activity, the precise structural mechanisms underlying these effects remain to be elucidated. In the revised discussion, we have attempted to provide a clearer framework for how Fe-S clusters might modulate RTC function, while acknowledging the current limitations in defining the exact nature of the conformational changes.

Specifically, our spectroscopic and functional assays indicate that redox transitions of the Fe-S clusters in nsp14 and nsp10 affect RNA binding and, in the case of nsp14 and the nsp10/nsp16 complex, methyltransferase activity, without altering the exoribonuclease function of the nsp14/nsp10 complex. These effects suggest a regulatory role for the clusters in RNA engagement rather than direct participation in catalysis. These observations suggest that the clusters may act as redox-responsive switches, modulating protein-RNA interactions and possibly influencing the dynamic assembly or progression of the RTC. These mechanisms are supported by analogies to well-characterized systems such as nitrogenase and respiratory complex I, where Fe-S clusters are known to mediate redox-driven conformational gating and electron transfer. Nevertheless, we emphasize that these ideas remain speculative and will require future structural and biophysical studies, such as cryo-EM comparisons of oxidized and reduced RTC complexes, or single-molecule FRET to monitor conformational dynamics, to be fully validated.

In summary, our findings highlight the presence of Fe-S clusters in the SARS-CoV-2 replication and transcription machinery, and provide a testable model for their mechanistic role. We have revised the discussion to clarify the current state of knowledge and to acknowledge the need for further investigation into the structural basis of Fe-S cluster-mediated regulation.

We hope that the substantial revisions, additional results, and clarifications provided in this revised manuscript have addressed the Reviewers' concerns and made the work suitable for publication in *Nature Communications*.

Sincerely,
Nunziata Maio, PhD
Tracey Rouault, MD

1. H. Borsook, G. Keighley, Oxidation-Reduction Potential of Ascorbic Acid (Vitamin C). *Proc Natl Acad Sci U S A* **19**, 875-878 (1933).
2. W. R. Hagen, *Biomolecular EPR spectroscopy*. (CRC Press, 2020).
3. T. V. Morgan *et al.*, Spectroscopic studies of ferricyanide oxidation of *Azotobacter vinelandii* ferredoxin I. *Proc Natl Acad Sci U S A* **81**, 1931-1935 (1984).
4. T. V. Morgan, P. J. Stephens, F. Devlin, B. K. Burgess, C. D. Stout, Selective oxidative destruction of iron-sulfur clusters. Ferricyanide oxidation of *Azotobacter vinelandii* ferredoxin I. *FEBS Lett* **183**, 206-210 (1985).

DEPARTMENT OF HEALTH & HUMAN SERVICES

Public Health Services
National Institutes of Health

Eunice Kennedy Shriver
National Institute of Child Health
And Human Development (NICHD)
Metals Biology and Molecular Medicine Group
Section on Human Iron Metabolism
Building 35A, Room 2D-824
35A Convent Drive, MSC 3758
Bethesda, MD 20892-3758, USA
T: 301-496-7060; F: 301-402-0078
E-mails: nunziata.maio@nih.gov;
rouault@mail.nih.gov

July 23rd, 2025

We thank the Reviewers for their time and constructive feedback throughout the review process. We appreciate their careful evaluation of our work.

Sincerely,
Nunziata Maio, PhD
Tracey Rouault, MD